# Generalized Protein Pocket Generation with Prior-Informed Flow Matching

**Zaixi Zhang**[1,2], **Marinka Zitnik**[3*], **Qi Liu**[1,2*]
1: School of Computer Science and Technology, University of Science and Technology of China
2:State Key Laboratory of Cognitive Intelligence, Hefei, Anhui, China
3:Harvard University
zaixi@mail.ustc.edu.cn, marinka@hms.harvard.edu, qiliuql@ustc.edu.cn

## Abstract

Designing ligand-binding proteins, such as enzymes and biosensors, is essential in bioengineering and protein biology. One critical step in this process involves designing protein pockets, the protein interface binding with the ligand. Current approaches to pocket generation often suffer from time-intensive physical computations or template-based methods, as well as compromised generation quality due to the overlooking of domain knowledge. To tackle these challenges, we propose PocketFlow, a generative model that incorporates protein-ligand interaction priors based on flow matching. During training, PocketFlow learns to model key types of protein-ligand interactions, such as hydrogen bonds. In the sampling, PocketFlow leverages multi-granularity guidance (overall binding affinity and interaction geometry constraints) to facilitate generating high-affinity and valid pockets. Extensive experiments show that PocketFlow outperforms baselines on multiple benchmarks, e.g., achieving an average improvement of 1.29 in Vina Score and 0.05 in scRMSD. Moreover, modeling interactions make PocketFlow a generalized generative model across multiple ligand modalities, including small molecules, peptides, and RNA.

## 1 Introduction

Proteins are the fundamental building blocks of living organisms, often interacting with ligands (e.g., small molecules, nucleic acids, and peptides) to execute their functions. Recently, computational methods have played critical roles in designing functional proteins binding with ligands with broad applications in bio-engineering and therapeutics [76, 52, 50, 51, 67, 8, 58]. For example, Polizzi et al., [65] leverage template-matching methods to design de novo proteins binding with the drug apixaban [15]; Yeh et al., [83] use deep learning methods to generate efficient light-emitting enzyme luciferases with selective substrate catalysis capabilities. To design such ligand-binding proteins, an essential step is to design protein pockets, the protein interface interacting with binding ligands [68, 39, 12, 28]. However, the complexity of ligand-protein interactions, the variability of protein sidechains, and sequence-structure relationships pose great challenges for pocket design [25, 51, 48].

Traditional methods for pocket design mainly focus on physics modeling or template-matching [12, 28, 65, 19, 62]. However, the involved physical energy calculation or substructure enumeration could be quite time-consuming. Recent advancements in pocket design have benefited a lot from deep learning-based approaches [73, 92, 83, 47, 51, 48]. However, these innovative approaches often overlook essential domain knowledge, such as the protein-ligand interactions and the geometric constraints governing them. Though they can efficiently generate many candidates, further screening/optimization is required to get valid and high-affinity pockets. Moreover, most methods are restricted to small molecule ligands, omitting other important ligand types such as nucleic acids [8] and peptides [53].

---

[*]Marinka Zitnik and Qi Liu are the corresponding authors.

38th Conference on Neural Information Processing Systems (NeurIPS 2024).

To tackle the aforementioned challenges, we propose PocketFlow, a protein-ligand interaction prior-informed flow matching model for protein pocket generation. Firstly, we define conditional flows for diverse data modalities in the protein-ligand complex including backbone frames, sidechain torsions, and residue/interaction types. We choose flow matching as the generative framework because of its efficiency and flexibility [17, 13, 57, 57]. Furthermore, PocketFlow explicitly learns the dominant protein-ligand interaction types including hydrogen bonds [35], salt bridges [24], hydrophobic interactions [61], and $\pi - \pi$ stacking [36], which are crucial for strong binding stability and affinity of protein-ligand pairs [2]. In the sampling process, binding affinity and interaction geometry guidance are adopted to encourage generating pockets with high affinity and validity. Specifically, we leverage a lightweight binding affinity predictor to predict the affinity of the generated complex and apply distance and angle constraints to promote desirable protein-ligand interactions. To circumvent the non-differentiability issues associated with residue type sampling, we employ a novel sidechain ensemble method for interaction geometry calculations. Extensive experiments show that PocketFlow provides a generalized framework for high-quality protein pocket generation across various ligand modalities (small molecules, RNA, peptides, etc.,). The code is provided at `https://github.com/zaixizhang/PocketFlow`. Our main contributions are summarized as:

- **Generalized tasks:** Our study broadens the scope of protein pocket generation tasks to include various ligand modalities such as small molecules, nucleic acids, and peptides.

- **Novel method:** PocketFlow combines the recent progress of flow-matching-based generative models and physical/chemical interaction priors (affinity guidance and interaction geometry guidance) to generate protein pockets with enhanced affinity and structural validity.

- **Strong performance:** PocketFlow outperforms existing methods on various benchmarks of pocket generation, producing an average improvement of 1.29 in Vina score and 0.05 in scRMSD. Further interaction analysis highlights the model's ability to foster beneficial protein-ligand interactions, e.g., an average of 4.12 hydrogen bonds, while markedly reducing steric clashes (an average of 1.21 in generated pockets v.s. 4.59 in the test set).

## 2 Related Works

### 2.1 Generative Models for Protein Generation

Recent advancements in deep generative models have significantly advanced the field of *de novo* protein structure generation, enabling researchers to create proteins with specific desired properties [81, 37, 84, 86, 13, 95, 94, 91]. For example, RFDiffusion [81] employs denoising diffusion probabilistic models [33] in conjunction with RoseTTAFold [7] for *de novo* protein structure generation. It achieved notable success by generating proteins validated in wet lab experiments. Chroma [37] leverages a similar diffusion process with efficient neural architecture for molecular systems that enables long-range reasoning with sub-quadratic scaling. It also demonstrates strong capabilities to satisfy constraints including symmetries, substructure, shape, semantics, and simple natural-language prompts. Recently, models leveraging flow matching frameworks have shown promising results on protein generation [86, 13, 85, 40, 53]. For example, FoldFlow [13] proposed a series of flow-matching-based generative models for protein backbones with improved training stability and efficiency than diffusion-based models. FrameFlow [84, 85] further shows sampling efficiency and achieves success on motif-scaffolding tasks with flow matching. However, these protein generation methods are not directly applicable to protein pocket generation that requires protein-ligand interaction modeling.

### 2.2 Protein Pocket Generation

Protein pockets are the protein interface where the ligand binds to the protein and pocket design is a critical task for bioengineering [68, 39, 12, 28]. Traditional methods for pocket design focus on physics modeling or template-matching [12, 28, 65, 19, 62, 93]. For example, PocketOptimizer [62] predicts mutations in protein pockets to increase binding affinity based on physical energy calculation, which may bring a large time burden. The recent progress in protein pocket design has been facilitated by deep generative models [73, 92, 83, 93, 48]. For instance, FAIR [92] co-designs pocket structures and sequences using a two-stage coarse-to-fine refinement approach. RFDiffusion All-Atom [48] extends RFDiffusion for joint modeling of protein and ligand structure to generate

ligand-binding protein and further leverages ProteinMPNN[21]/LigandMPNN[22] for sequence design. However, deep-learning methods lacking physical/chemical prior guidance may be less accurate and generalizable. In PocketFlow, we aim to design prior-guided pocket generative models.

# 3 Preliminaries

## 3.1 Notations and Problem Formulation

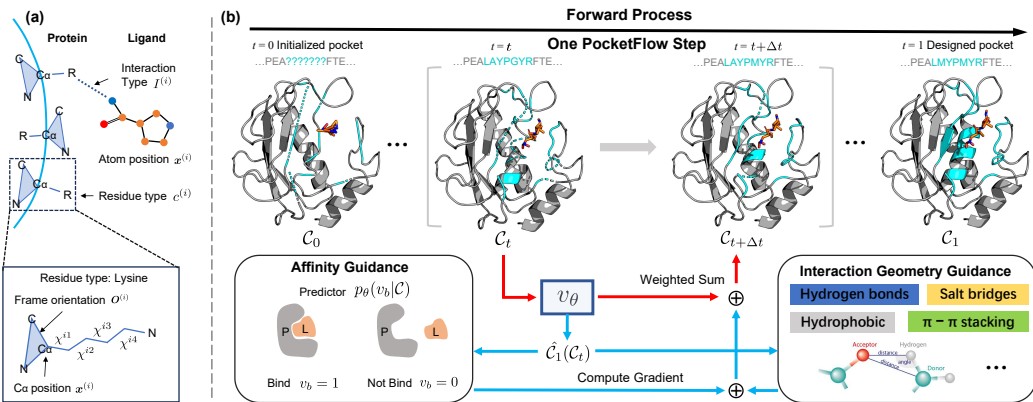

Figure 1: (a) Parameterization of protein-ligand complex. (b) Illustration of PocketFlow forward process. The affinity and interaction geometry guidance are proposed to improve affinity and structural validity. The red/blue lines denote the unconditional/guidance paths respectively.

**Notations.** As shown in Figure 1(a), we model protein-ligand complex as $\mathcal{C} = \{\mathcal{P}, \mathcal{G}\}$ consisting of protein $\mathcal{P}$ and ligand $\mathcal{G}$ (small molecule as an example). Protein $\mathcal{P}$ is composed of a sequence of residues (amino acids) with residue types denoted $c^{(i)} \in \mathbb{R}^{20}$. Consistent with [92, 87], the protein pocket $\mathcal{R} \subset \mathcal{P}$ is defined as the subset of residues closest to the ligand atoms under a threshold $\delta$ (e.g., 3.5 Å). In a residue, the backbone structure (consisting of $C_\alpha$, $N$, $C$, $O$) is parameterized with $C_\alpha$ position $x^{(i)} \in \mathbb{R}^3$ and a frame orientation matrix $O^{(i)} \in SO(3)$ following [43, 84]. The sidechain is parameterized with maximal 4 torsion angles $\chi^{(i)} = \{\chi^{i1}, \chi^{i2}, \chi^{i3}, \chi^{i4}\} \in [0, 2\pi)^4$. Given these key parameters, the full atom protein structure can be derived with the ideal frame coordinates and the sidechain bond length/angles [43]. The protein-ligand interaction type for each residue is marked as $I^{(i)} \in \mathbb{R}^5$ (Hydrogen bond, Salt bridge, Hydrophobic, $\pi$-$\pi$ stacking, no interaction). A pocket with $N_r$ residues can be compactly represented as $\mathcal{R} = \{c^{(i)}, x^{(i)}, O^{(i)}, \chi^{(i)}, I^{(i)}\}_{i=1}^{N_r}$. As for the ligand, we use a generalized atom-level representation that accommodates various modalities including small molecules, peptides, and RNA. The atom types and bonding information between atoms are given and PocketFlow predicts the $N_l$ ligand atom coordinates (also denoted as $x^{(i)}$ for conciseness).

**Problem Formulation.** PocketFlow co-designs residue types and 3D structures of the protein pocket conditioned on the ligand (could be small molecules, nucleic acids, peptides, etc.) and protein scaffold (the other parts of protein besides the pocket region, i.e., $\mathcal{P} \setminus \mathcal{R}$). The ligand structure $\mathcal{G}$ is also predicted. Formally, PocketFlow aims to learn a conditional generative model $p_\theta(\mathcal{R}, \mathcal{G}|\mathcal{P} \setminus \mathcal{R})$.

## 3.2 Preliminaries on Flow Matching

Flow matching (FM) [57] is a simulation-free method for learning continuous normalizing flows (CNFs) [18] that generates data by integrating an ordinary differential equation (ODE) over a learned vector field. Here, we first give an overview of Riemannian flow matching [17]. On a manifold $\mathcal{M}$, the CNF $\psi_t(\cdot): \mathcal{M} \to \mathcal{M}$ is defined by integrating along a time-dependent vector field $u_t(x) \in \mathcal{T}_x\mathcal{M}$ where $\mathcal{T}_x\mathcal{M}$ denotes the tangent space of the manifold at $x \in \mathcal{M}$: $\frac{d}{dt}\psi_t(x) = u_t(\psi_t(x)), \psi_0(x) = x, t \in [0, 1]$. The flow transforms a simple distribution $p_0$ towards the data distribution $p_1$. In FM, the target is to learn a neural network $v_\theta(x, t)$ that approximates $u_t(x)$, while the vanilla regression loss: $\mathcal{L}_{FM}(\theta) = \mathbb{E}_{t \sim \mathcal{U}[0,1], p_t(x)} \|v_\theta(x, t) - u_t(x)\|_g^2$ is hard to compute in practice. Here, $\mathcal{U}[0, 1]$ is the uniform distribution between 0 and 1, and $\| \cdot \|_g^2$ is the norm induced by the Riemannian

metric $g$. Instead, it is tractable to define conditional vector field $u_t(x|x_1)$ and obtain the conditional FM objective: $\mathcal{L}_{CFM}(\theta) = \mathbb{E}_{t \sim \mathcal{U}[0,1], p_1(x_1), p_t(x|x_1)} \|v_\theta(x,t) - u_t(x|x_1)\|_g^2$. It has been proved that $\nabla_\theta \mathcal{L}_{FM}(\theta) = \nabla_\theta \mathcal{L}_{CFM}(\theta)$ [57, 17]. In the inference, ODE solvers are applied to solve the ODE, e.g., $x_1 = \text{ODESolve}(x_0, v_\theta, 0, 1)$ where $x_0$ is the initialized data and $x_1$ is the generated data.

# 4 PocketFlow

PocketFlow is an interaction prior-informed flow-matching model for pocket design. In this section, we first define PocketFlow for different components in the protein-ligand complex (backbone in Sec. 4.1, sidechain in Sec. 4.2, and residue/interaction types in Sec.4.3). Then we show the prior-informed training and sampling in Sec. 4.4 and 4.5.

## 4.1 PocketFlow on SE(3)

As introduced in Sec. 3.1, each residue frame can be parameterized by a rigid transformation $T = (\boldsymbol{x}^{(i)}, \boldsymbol{O}^{(i)})$ within SE(3) space. The backbone with $N_r$ residues can thus be described by a set of transformations $[T^{(1)}, \ldots, T^{(N_r)}]$ belonging to SE(3)$^{N_r}$ and constitutes a product space. The following deduction focuses on a single frame but can be generalized to the whole protein backbone. The $C_\alpha$ coordinates $\boldsymbol{x}^{(i)}$ are initialized with linear interpolation and extrapolation based on the coordinates of neighboring scaffold residues following [92]. The prior distribution of $\boldsymbol{O}^{(i)}$ is chosen as the uniform distribution on SO(3). Following previous works [17, 84], the conditional flow for $\boldsymbol{x}^{(i)}$ and $\boldsymbol{O}^{(i)}$ are defined as $\boldsymbol{x}_t^{(i)} = (1-t)\boldsymbol{x}_0^{(i)} + t\boldsymbol{x}_1^{(i)}$ and $\boldsymbol{O}_t^{(i)} = \exp_{\boldsymbol{O}_0^{(i)}}(t \log_{\boldsymbol{O}_0^{(i)}}(\boldsymbol{O}_1^{(i)}))$ respectively, which are geodesic paths in $\mathbb{R}^3$ and SO(3). The exponential map $\exp_{\boldsymbol{O}_0}$ can be computed using Rodrigues' formula and the logarithmic map $\log_{\boldsymbol{O}_0}$ is similarly easy to compute with its Lie algebra $\mathfrak{so}(3)$ [84]. The loss function of PocketFlow on SE(3) is the summation of the two losses below:

$$\mathcal{L}_{coord}(\theta) = \mathbb{E}_{t, p_1(\boldsymbol{x}_1), p_0(\boldsymbol{x}_0), p_t(\boldsymbol{x}_t|\boldsymbol{x}_0, \boldsymbol{x}_1)} \frac{1}{N_r + N_l} \sum_{i=1}^{N_r + N_l} \left\| v_\theta^{(i)}(\boldsymbol{x}_t^{(i)}, t) - \boldsymbol{x}_1^{(i)} + \boldsymbol{x}_0^{(i)} \right\|_2^2, \quad (1)$$

$$\mathcal{L}_{ori}(\theta) = \mathbb{E}_{t, p_1(\boldsymbol{O}_1), p_0(\boldsymbol{O}_0), p_t(\boldsymbol{O}_t|\boldsymbol{O}_0, \boldsymbol{O}_1)} \frac{1}{N_r} \sum_{i=1}^{N_r} \left\| v_\theta^{(i)}(\boldsymbol{O}_t^{(i)}, t) - \frac{\log_{\boldsymbol{O}_t^{(i)}}(\boldsymbol{O}_1^{(i)})}{1-t} \right\|_{SO(3)}^2, \quad (2)$$

where we additionally consider $N_l$ ligand atom coordinates in $\mathcal{L}_{coord}(\theta)$, for which we use Gaussian distribution at the center of ligand mass as the prior distribution.

## 4.2 PocketFlow on Torus

As described in Sec. 3.1, the sidechain conformation of each residue can be represented as maximally four torsion angles $\boldsymbol{\chi}^{(i)} = \{\chi^{i1}, \chi^{i2}, \chi^{i3}, \chi^{i4}\} \in [0, 2\pi)^4$. In a pocket with $N_r$ residues, the sidechain torsion angles form a hypertorus $\mathbb{T}^{4N_r}$, which is the quotient space $\mathbb{R}^{4N_r}/2\pi\mathbb{Z}^{4N_r}$ with the equivalence relation: $\boldsymbol{\chi} = (\chi^1, \ldots, \chi^{4N_r}) \sim (\chi^1 + 2\pi, \ldots, \chi^{4N_r}) \sim (\chi^1, \ldots, \chi^{4N_r} + 2\pi)$ [41, 90]. Following [42], the prior distribution is chosen as a uniform distribution over $\mathbb{T}^{4N_r}$. We regard the torsion angles as mutually independent and use interpolation paths as: $\boldsymbol{\chi}_t = (1-t)\boldsymbol{\chi}_0 + t(\boldsymbol{\chi}_1' - \boldsymbol{\chi}_0)$ where $\boldsymbol{\chi}_1' = (\boldsymbol{\chi}_1 - \boldsymbol{\chi}_0 + \pi) \mod (2\pi) - \pi + \boldsymbol{\chi}_0$. The loss for the torsion angles is defined as:

$$\mathcal{L}_{tor}(\theta) = \mathbb{E}_{t, p_1(\boldsymbol{\chi}_1), p_0(\boldsymbol{\chi}_0), p_t(\boldsymbol{\chi}_t|\boldsymbol{\chi}_0, \boldsymbol{\chi}_1)} \frac{1}{N_r} \sum_{i=1}^{N_r} \left\| v_\theta^{(i)}(\boldsymbol{\chi}_t^{(i)}, t) - \boldsymbol{\chi}_1'^{(i)} + \boldsymbol{\chi}_0^{(i)} \right\|_2^2. \quad (3)$$

## 4.3 PocketFlow on Residue Types and Interaction Types

Each residue is assigned a probability vector with 20 dimensions: $\boldsymbol{c}^{(i)} \in \mathbb{R}^{20}$. The prior distribution is set as the uniform distribution and the conditional flow is defined as the Euclidean interpolation between $\boldsymbol{c}_0$ and $\boldsymbol{c}_1$ (one hot vector indicating residue type). $\boldsymbol{c}_t$ is a probability vector because its summation over all types equals 1. We leverage the cross-entropy loss $\text{CE}(\cdot, \cdot)$ following [53, 73, 16]:

$$\mathcal{L}_{res} = \mathbb{E}_{t \sim \mathcal{U}(0,1), p_1(\boldsymbol{c}_1), p_0(\boldsymbol{c}_0), p_t(\boldsymbol{c}|\boldsymbol{c}_0, \boldsymbol{c}_1)} \sum_{i=1}^{N_r} \text{CE}\left( \boldsymbol{c}_t^{(i)} + (1-t)v_\theta^{(i)}(\boldsymbol{c}_t^{(i)}, t), \boldsymbol{c}_1^{(i)} \right), \quad (4)$$

which measures the difference between the true probability and the inferred one $\hat{c}_1^{(i)} = c_t^{(i)} + (1 - t)v_\theta^{(i)}(c_t^{(i)}, t)$. We also note the recent progress of the sequential flow matching methods [74, 16], which can be seamlessly integrated into PocketFlow and are left for future works.

It has been shown that modeling **Protein-ligand interactions** explicitly in biomolecular generative models can effectively enhance the generalizability [89, 97]. We used the protein–ligand interaction profiler (PLIP) [69] to detect and annotate the protein-ligand interactions for each residue by analyzing their binding structure. Following [97], 4 dominant interactions are considered including salt bridges, $\pi$–$\pi$ stacking, hydrogen bonds, and hydrophobic interactions. For simplicity, if a residue has more than one interaction, we take the one with the highest rank, which considers both the contribution to the binding affinity and the frequencies (see Appendix. B). Similar to residue types, interactions are modeled as category data: $I = \{I^{(i)}\}_{i=1}^{N_r}$. Besides the 4 interaction types, we also consider an unknown/none type. Similar to Equ. 4, we have the interaction loss:

$$\mathcal{L}_{inter} = \mathbb{E}_{t \sim \mathcal{U}(0,1), p_1(I_1), p_0(I_0), p_t(I|I_0, I_1)} \sum_{i=1}^{N_r} \text{CE}\left(I_t^{(i)} + (1-t)v_\theta^{(i)}(I_t^{(i)}, t), I_1^{(i)}\right). \quad (5)$$

### 4.4 Model Training

**Network Architecture.** To design the binding protein pocket $\mathcal{R} = \{c^{(i)}, x^{(i)}, O^{(i)}, \chi^{(i)}, I^{(i)}\}_{i=1}^{N_r}$ and update the binding ligand coordinates $\{x^{(i)}\}_{i=1}^{N_l}$, we utilize an architecture modified from the FrameDiff [86] which incorporates Invariant Point Attention (IPA) from AF2 [43] to encode spatial features combined with transformer layers [79] to encode sequence-level features. To achieve a unified representation of both protein residues and ligand atoms, we follow the approach used in RoseTTAFold All-Atom [49], where each ligand atom is treated as an individual residue. Initial representations are based on atom element type embeddings, and the frame orientations are set as identity matrices. To further model the covalent bonding information (single bond, double bond, triple bond, or aromatic bond), we also add the bond embeddings to the 2D track. We use additional MLPs based on the residue embeddings to predict the residue types, interaction types, and sidechain torsion angles. Instead of directly predicting the vector field, we let the model predict the final structure at $t = 1$ and interpolate to obtain the vector field. More details are introduced in the Appendix. C.

**Overall Training Loss.** The overall training loss of PocketFlow is the summation of Equ. 1, 2, 3, 4, and 5. To fully utilize the protein-ligand context information, we use the whole protein-ligand complex structure at $t$, i.e., $\mathcal{C}_t = \mathcal{P}_t \cup \mathcal{G}_t$ as the inputs of $v_\theta(\cdot, t)$.

**Equivariance.** Following [84, 53], we perform all training and sampling within the zero center of mass (CoM) subspace by subtracting the CoM of the scaffold from the initialized structure. PocketFlow has the ideal SE(3)-equivariance property of geometric generative models:

**Theorem 1.** *Denote the SE(3)-transformation as $T_g$, PocketFLow $p_\theta(\mathcal{R}, \mathcal{G}|\mathcal{P} \setminus \mathcal{R})$ is SE(3) equivariant i.e., $p_\theta(T_g(\mathcal{R}, \mathcal{G})|T_g(\mathcal{P} \setminus \mathcal{R})) = p_\theta(\mathcal{R}, \mathcal{G}|\mathcal{P} \setminus \mathcal{R})$, where $\mathcal{R}$ denotes the designed pocket, $\mathcal{G}$ is the binding ligand, and $\mathcal{P} \setminus \mathcal{R}$ is the protein scaffold.*

The main idea is that the SE(3)-invariant prior and SE(3)-equivariant neural network lead to an SE(3)-equivariant generative process of PocketFlow. We give the full proof in the Appendix. D.

### 4.5 Pocket Sampling with Prior Guidance

To improve the binding affinity and structural validity of the generated protein pocket, we proposed a novel domain-knowledge-guided sampling scheme. Generally, we use classifier-guided sampling [23] and consider overall binding affinity guidance and interaction geometry guidance. To encourage the generated protein-ligand complex to satisfy a specific condition $y$, we apply the Bayes rule [23, 30]:

$$\nabla_{\mathcal{C}_t} \log p(\mathcal{C}_t|y) = \nabla_{\mathcal{C}_t} \log p(\mathcal{C}_t) + \nabla_{\mathcal{C}_t} \log p(y|\mathcal{C}_t), \quad (6)$$

where $\nabla_{\mathcal{C}_t} \log p(\mathcal{C}_t)$ is the unconditional vector field $v_\theta(\mathcal{C}_t, t)$ and $\nabla_{\mathcal{C}_t} \log p(y|\mathcal{C}_t)$ is the guidance term to constrain the generated complex in a specific condition $y$.

**Binding Affinity Guidance.** To generate protein pockets with higher binding affinity to the target ligand, we train a separate lightweight affinity predictor for guidance (More details of the predictor in Appendix. E.1). Specifically, the data points in the training set are annotated 1 if their affinity is

higher than the average score of the dataset, otherwise 0 [66]. Because the intermediate structure is noisy, we take the expected structure at $t = 1$, i.e., $\hat{\mathcal{C}}_1(\mathcal{C}_t)$ from the model output and feed it into the predictor. Then we have the classifier-guided velocity field $\tilde{v}_\theta(\mathcal{C}_t, t)$:

$$\tilde{v}_\theta(\mathcal{C}_t, t) = v_\theta(\mathcal{C}_t, t) - \gamma \nabla_{\mathcal{C}_t} \log p_\theta(v_b = 1 | \hat{\mathcal{C}}_1(\mathcal{C}_t)), \tag{7}$$

where we add a scaling factor $\gamma > 0$ that controls the gradient strength. $p_\theta$ is the affinity predictor and $v_b \in \{0, 1\}$ is the binary label of binding affinity.

**Interaction Geometry Guidance.** Inspired by [97, 89], we considered 4 dominant non-covalent interaction types in PocketFlow, including salt bridges, $\pi$–$\pi$ stacking, hydrogen bonds, and hydrophobic interactions. The local geometries need to satisfy a series of distance/angle constraints to form strong interactions [69]. For example, for hydrogen bonds, the distances between donor and acceptor atoms need to be less than 4.1 Å and larger than 2 Å to reduce steric clashes [35]. The following inequality is a necessary condition for residues in $\hat{\mathcal{C}}_1(\mathcal{C}_t)$ with predicted interaction label $\hat{I}_1$ as hydrogen bond:

$$l_{\min} \leq \min_{i \in \mathcal{A}_{hbond}^{(k)}, j \in \mathcal{G}} \left\| \boldsymbol{x}^{(i)} - \boldsymbol{x}^{(j)} \right\|_2 \leq l_{\max}, \tag{8}$$

where $l_{\min}$ and $l_{\max}$ are distance constraints; $\mathcal{A}_{hbond}^{(k)}$ denote the $k$-th residue in the set of residues with predicted hydrogen bonds. With a little abuse of notations, $\boldsymbol{x}^{(i)}$ and $\boldsymbol{x}^{(j)}$ denote the candidate atom coordinates in the residue and ligand respectively. The distance guidance can be derived as:

$$-\nabla_{\mathcal{C}_t} \sum_{k=1}^{|\mathcal{A}_{hbond}|} \left[ \xi_1 \max \left( 0, d^{(k)} - l_{\max} \right) + \xi_2 \max \left( 0, l_{\min} - d^{(k)} \right) \right], \tag{9}$$

where $d^{(k)} = \min_{i \in \mathcal{A}_{hbond}^{(k)}, j \in \mathcal{G}} \left\| \boldsymbol{x}^{(i)} - \boldsymbol{x}^{(j)} \right\|_2$ and $\xi_1, \xi_2 > 0$ are constant coefficients that control the strength of guidance. The detailed deduction is included in the Appendix. E.2. Besides the distance constraint, the hydrogen bond needs to satisfy the acceptor/donor angle constraint [69], e.g., the donor/acceptor angle needs to be larger than $100°$. The angle guidance is presented as follows:

$$-\xi_3 \nabla_{\mathcal{C}_t} \sum_{k=1}^{|\mathcal{A}_{hbond}|} \max(0, \alpha_{\min} - \phi^{(k)}), \tag{10}$$

where $\phi^{(k)} = \max_{i \in \mathcal{A}_{hbond}^{(k)}, j \in \mathcal{G}}$ hangle$(\boldsymbol{x}^{(i)}, \boldsymbol{x}^{(j)})$ and hangle$(\cdot, \cdot)$ calculates the acceptor/donor angle in Figure. 4. $\xi_3 > 0$ is the guidance coefficient. The guidance for the other interactions is discussed in Appendix. E. We note that the residue type/side chain structure of the pocket is not determined during the sampling. Directly sampling from the residue type distribution makes the model not differentiable [38]. We propose the **Sidechain Ensemble** for the interaction geometry calculation, i.e., the weighted sum of geometric guidance with respect to residue types (Figure. 6).

**Sampling.** With the initialized data, the sampling process is the integration of the ODE $\frac{d\mathcal{C}_t}{dt} = v_\theta(\mathcal{C}_t, t)$ from $t = 0$ to $t = 1$ with an Euler solver [14]. $\gamma, \xi_1, \xi_2$, and $\xi_3$ are set as 1 in the default setting. To apply the guidance, we use $\tilde{v}_\theta$ which is $v_\theta$ plus guidance terms (Equ. 7, 9, and 10):

$$\boldsymbol{\chi}_{t+\Delta t}^{(i)} = \text{reg} \left( \boldsymbol{\chi}_t^{(i)} + \tilde{v}_\theta(\boldsymbol{\chi}_t^{(i)}, t)\Delta t \right); \tag{11}$$

$$\boldsymbol{x}_{t+\Delta t}^{(i)} = \boldsymbol{x}_t^{(i)} + \tilde{v}_\theta(\boldsymbol{x}_t^{(i)}, t)\Delta t; \quad \boldsymbol{O}_{t+\Delta t}^{(i)} = \boldsymbol{O}_t^{(i)} \exp \left( \tilde{v}_\theta(\boldsymbol{O}_t^{(i)}, t)\Delta t \right); \tag{12}$$

$$\boldsymbol{c}_{t+\Delta t}^{(i)} = \text{norm} \left( \boldsymbol{c}_t^{(i)} + \tilde{v}_\theta(\boldsymbol{c}_t^{(i)}, t)\Delta t \right); \quad I_{t+\Delta t}^{(i)} = \text{norm} \left( I_t^{(i)} + \tilde{v}_\theta(I_t^{(i)}, t)\Delta t \right); \tag{13}$$

where $\Delta t$ is the time step; $v_\theta(\cdot; t)$ denotes the subcomponent of the vector field for different variables. norm$(\cdot)$ means normalizing the vector to a probability vector such that the summation is 1, and reg$(\cdot)$ means regularizing the torsion angles by reg$(\tau) = (\tau + \pi) \mod (2\pi) - \pi$.

## 5 Experiments

### 5.1 Experimental Settings

**Datasets.** Following previous works [29, 71, 92] we consider two widely used protein-small molecule binding datasets for experimental evaluations: **CrossDocked** dataset [27] is generated through cross-docking and is split with mmseqs2 [75] at 30% sequence identity, leading to train/val/test set of

Table 1: Evaluation of different models on **small-molecule-binding** protein pocket design. We report the average and standard deviation values across three independent runs. We highlight the best two results with **bold text** and underlined text, respectively.

| Model | CrossDocked | | | Binding MOAD | | |
|---|---|---|---|---|---|---|
| | AAR (↑) | scRMSD (↓) | Vina (↓) | AAR (↑) | scRMSD (↓) | Vina (↓) |
| Test set | - | 0.65 | -7.016 | - | 0.67 | -8.076 |
| DEPACT | 31.52±3.26% | 0.73±0.06 | -6.632±0.18 | 35.30±2.19% | 0.77±0.08 | -7.571±0.15 |
| dyMEAN | 38.71±2.16% | 0.79±0.09 | -6.855±0.06 | 41.22±1.40% | 0.80±0.12 | -7.675±0.09 |
| FAIR | 40.16±1.17% | 0.75±0.03 | -7.015±0.12 | 43.68±0.92% | 0.72±0.04 | -7.930±0.15 |
| RFDiffusionAA | 50.85±1.85% | 0.68±0.07 | -7.012±0.09 | 49.09±2.49% | 0.70±0.04 | -8.020±0.11 |
| PocketFlow | **52.19±1.34%** | 0.67±0.04 | **-8.236±0.16** | **54.30±1.70%** | 0.68±0.03 | **-9.370±0.24** |
| w/o Aff Guide | 50.94±1.37% | **0.65±0.04** | -7.375±0.10 | 51.43±1.52% | **0.64±0.04** | -8.380±0.19 |
| w/o Geo Guide | 49.80±1.41% | 0.68±0.03 | -8.120±0.14 | 53.49±1.53% | 0.71±0.05 | -9.197±0.22 |
| w/o Geo & Aff Guide | 48.50±1.66% | 0.71±0.06 | -7.135±0.13 | 49.71±1.68% | 0.69±0.03 | -8.241±0.18 |
| w/o Inter Learning | 50.72±1.20% | 0.66±0.03 | -7.968±0.15 | 52.25±1.74% | 0.68±0.05 | -9.031±0.17 |

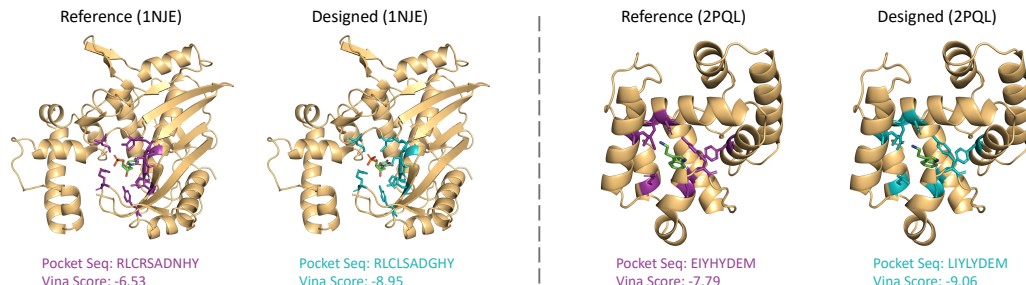

Reference (1NJE)    Designed (1NJE)    Reference (2PQL)    Designed (2PQL)

Pocket Seq: RLCRSADNHY   Pocket Seq: RLCLSADGHY   Pocket Seq: EIYHYDEM   Pocket Seq: LIYLYDEM
Vina Score: -6.53   Vina Score: -8.95   Vina Score: -7.79   Vina Score: -9.06

Figure 2: Case studies of small-molecule-binding protein pocket design. We show the reference and designed structures/sequences of two protein pockets from the CrossDocked (PDB ID: 1NJE) and Binding MOAD (PDB ID: 2PQL) datasets respectively.

100k/100/100 complexes. **Binding MOAD** dataset [34] contains experimentally determined protein-small molecule complexes and is split based on the proteins' enzyme commission number [9], resulting in 40k protein-small molecule pairs for training, 100 pairs for validation, and 100 pairs for testing. To test the generalizability of PocketFlow to other ligand modalities, we further consider **PPDBench** [3], which contains 133 non-redundant complexes of protein-peptides and **PDBBind RNA** [80], which contains 56 protein-RNA pairs by filtering the PDBBind nucleic acid subset. More details of data preprocessing are included in the Appendix. A. Considering the distance ranges of protein-ligand interactions [60], we redesign all the protein residues that contain atoms within 3.5 Å of any binding ligand atoms, i.e., the pocket area. The number of designed residues is set the same as the reference pocket. We sample 100 pockets for each complex in the test set for evaluation.

**Baselines and Implementation.** PocketFlow is compared with four state-of-the-art representative baseline methods. **DEPACT** [19] is a template-matching method [88] for pocket design by searching and grafting residues. **dyMEAN** [47] and **FAIR** [92] are deep-learning methods based on equivariant translation and iterative refinement. **RFDiffusionAA** [48] is the latest version of RFDiffusion [81], which can directly generate protein pocket structures conditioned on the ligand. Different from the sequence-structure co-design scheme in dyMEAN and FAIR, the residue types and sidechain structures in RFDiffusionAA are determined with LigandMPNN [22] in a post-hoc manner.

For experiments on PPDBench and PDBBind RNA, we pretrain PocketFlow and baseline models on the combination of the CrossDocked and Binding MOAD dataset, in which we carefully eliminate all complexes with the same PDB IDs to avoid potential data leakage. Then we represent the peptide and RNA similar to small molecules (atom coordinates/types and covalent bonds) and input to PocketFlow for sampling without fine-tuning. For simplicity, the structure of peptide/RNA is set fixed. All the baselines are run on the same Tesla A100 GPU.

**Performance Metrics.** We employ the following metrics to comprehensively evaluate the validity of the generated pocket sequence and structure. Amino Acid Recovery (**AAR**) is defined as the overlapping ratio between the predicted and ground truth residue types. In bioengineering, mutating too many residues may lead to instability or failure to fold [4]. Therefore, a larger AAR is more favorable. **scRMSD** refers to self-consistency Root Mean Squared Deviation between the generated and the predicted pocket's backbone atoms to reflect structural validity. Following established pipelines

Table 2: Evaluation of different approaches on the **peptide** and **RNA** datasets. DEPACT is not reported here because it is specially designed for small molecules. dyMEAN, FAIR, and PocketFlow are pretrained on protein-small molecule datasets and we use the checkpoint of RFDiffusionAA [1].

| Model | PPDBench | | | PDBBind RNA | | |
|---|---|---|---|---|---|---|
| | AAR ($\uparrow$) | scRMSD ($\downarrow$) | $\Delta\Delta G$ ($\downarrow$) | AAR ($\uparrow$) | scRMSD ($\downarrow$) | $\Delta\Delta G$ ($\downarrow$) |
| Test set | - | 0.64 | - | - | 0.59 | - |
| dyMEAN | 26.29±1.05% | 0.71±0.05 | -0.23±0.04 | 25.90±1.22% | 0.71±0.04 | -0.18±0.03 |
| FAIR | 32.53±0.89% | 0.86±0.04 | 0.05±0.07 | 24.90±0.92% | 0.80±0.05 | 0.13±0.05 |
| RFDiffusionAA | 46.85±1.45% | **0.65±0.06** | -0.62±0.05 | **44.69±1.90%** | **0.65±0.03** | -0.45±0.07 |
| PocketFlow | **48.19±1.34%** | 0.67±0.04 | **-1.06±0.04** | 44.34±1.16% | 0.69±0.01 | **-0.78±0.07** |
| w/o Aff Guide | 47.78±1.18% | 0.70±0.02 | -0.47±0.10 | 42.15±1.56% | 0.68±0.04 | -0.35±0.11 |
| w/o Geo Guide | 47.30±1.94% | 0.72±0.05 | -0.96±0.08 | 41.73±2.34% | 0.77±0.09 | -0.65±0.15 |
| w/o Geo & Aff Guide | 44.63±1.79% | 0.78±0.05 | -0.31±0.05 | 39.70±1.24% | 0.78±0.06 | -0.26±0.08 |
| w/o Inter Learning | 36.41±1.38% | 0.74±0.06 | -0.34±0.05 | 36.27±1.47% | 0.82±0.13 | -0.23±0.06 |

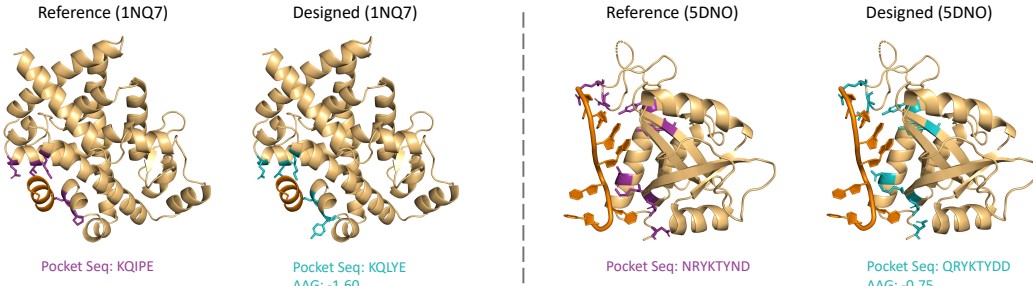

Figure 3: Case studies of peptide/RNA-binding protein pocket design. We show the reference and designed structures/sequences of two protein pockets from PPDBench (PDB ID: 1NQ7) and PDBBind RNA (PDB ID: 5DNO) datasets respectively. The ligand structures (orange) are set fixed.

[77, 54], for each generated protein structure, eight sequences are firstly derived by ProteinMPNN [21] and the folded to structures with ESMFold [56]. we report the minimum scRMSD for the predicted structures. To measure the binding affinity for protein-small molecule pairs, we calculate **Vina Score** with AutoDock Vina [78] following [64, 92]. For protein-peptide and protein-RNA pairs, we calculate **Rosetta $\Delta\Delta G$** [5] and **Rosetta-Vienna RNP-$\Delta\Delta G$** [44] respectively that measure the binding affinity change. The unit is kcal/mol and a lower Vina score/$\Delta\Delta G$ indicates higher affinity.

## 5.2 Small-molecule-binding Pocket Design

Table 1 shows the results of different methods on the CrossDocked and Binding MOAD dataset for small-molecule-binding pocket design. We can observe that PocketFlow overperforms baseline models with a clear margin on AAR, scRMSD, and Vina scores, demonstrating the strong ability of PocketFlow to design pockets with high validity and affinity. The average improvements over the RFDiffusionAA on AAR, scRMSD, and Vina Score are 3.3%, 0.05, and 1.29 respectively. PocketFlow also predicts more aligned sidechain angles with ground truth as evidenced in Table. 5. Compared with baselines, PocketFlow enjoys the advantage of powerful flow-matching architecture and effective physical/chemical prior guidance. Different from the post-hoc manner of deriving sequences in RFDiffusionAA [48], the co-design scheme also encourages sequence-structure consistency. We also compare the *generation efficiency* of different models in Figure. 7. Considering the pocket quality improvement brought by PocketFlow, the time overhead is acceptable.

We also perform ablation studies in Table. 1, where w/o Aff Guide, w/o Geo Guide, and w/o Geo & Aff Guide indicate generating pockets without Affinity Guidance, Interaction Geometry Guidance, and all Guidance respectively. In w/o Inter Learning, we retrain a model without learning interaction types and generate pockets without Interaction Geometry Guidance as well. We can observe that the Affinity and Geometry Guidance indeed play critical roles in enhancing binding affinity and structural validity. For example, the Vina score drops to -7.135 without guidance from -8.236 on the CrossDocked dataset. We also note Affinity Guidance may have slight side effects on scRMSD and we need to balance the strength of unconditional and guidance terms (more results in Appendix. F).

| Methods | Clash ($\downarrow$) | HB ($\uparrow$) | Salt ($\uparrow$) | Hydro ($\uparrow$) | $\pi$-$\pi$ ($\uparrow$) |
|---|---|---|---|---|---|
| Test set | 4.59 | 3.89 | 0.26 | 5.89 | **0.32** |
| DEPACT | 6.72 | 3.10 | 0.14 | 5.70 | 0.16 |
| dyMEAN | 4.65 | 3.07 | 0.17 | 5.85 | 0.20 |
| FAIR | 4.90 | 3.30 | 0.18 | 5.47 | 0.15 |
| RFDiffusionAA | 3.58 | 3.76 | 0.22 | 5.65 | 0.31 |
| PocketFlow | **1.21** | **4.12** | **0.27** | **6.03** | 0.28 |

Table 3: Interaction analysis of the generated protein pockets on the CrossDocked dataset. We measure the average number of steric clashes (**Clash**), hydrogen bonds (**HB**), salt bridges (**Salt**), hydrophobic interactions (**Hydro**), and $\pi$-$\pi$ stacking ($\pi - \pi$) per protein-ligand complex. More results on the variants of PocketFlow are included in Appendix F.

### 5.3 Generalization to Other Ligand Domains

Besides small molecules, the binding of protein with other ligand modalities such as peptides and nucleic acids play critical roles in biomedicine [82, 8]. However, the available dataset size compared with small molecules-protein complexes is quite limited (e.g., $\sim 100$ in PPDBench v.s. over 100k in CrossDocked). Here, we explore whether the pretrained PocketFlow on the combination of CrossDocked and Binding MOAD can generalize to peptide and RNA-binding pocket design in Table. 2. The peptide and RNA ligands are represented as molecules (atoms and covalent bonds) to fit into the pretrained models. We have observed that PocketFlow achieves performance comparable to the state-of-the-art baseline, RFDiffusionAA, with prior guidance significantly enhancing its generalizability. Our hypothesis is that the protein-ligand interactions and fundamental physical laws learned by PocketFlow are applicable universally across various biomolecular domains [89, 97]. By explicitly incorporating physical and chemical priors into the generative model, PocketFlow not only aligns with these universal principles but also gains a marked advantage of generalizability.

### 5.4 Interaction Analysis and Case Studies

We adopt PLIP [69] and posecheck [31] to detect the protein-ligand interactions in the generated pockets. In Table. 3, we show the average number of steric clashes, hydrogen bond donors, acceptors, and hydrophobic interactions (without redocking). We observe that PocketFlow can generate pockets with fewer clashes and more favorable interactions. For example, the average steric clashes for RFDiffusionAA and PocketFlow are 3.58 and 1.21 respectively. The average number of Hydrogen Bonds for RFDiffusionAA and PocketFlow are 3.76 and 4.12 respectively. These improvements can be attributed to the model's affinity/geometry guidance and its enhanced modeling of pocket/ligand flexibility, both of which promote the formation of advantageous protein-ligand interactions while minimizing clashes. Some interaction types such as $\pi$-$\pi$ stacking in PocketFlow are a little less than the reference, which may be due to the low frequency of these interactions in the dataset.

Figure. 2 and 3 show examples of the generated pockets for small molecules, peptides, and RNA. PocketFlow recovers most residue types and changes several key residues to achieve higher binding affinity. The overall structure of the pocket, including the sidechains, is generally well-maintained.

### 5.5 Limitations and Broader Impacts

While PocketFlow is a powerful generative method for pocket generation, we find the following limitations for further improvement. First, PocketFlow is only trained on protein-small molecule datasets in the paper. In the future, incorporating protein-peptides/nucleic acids/metal datasets, even the generated data from AlphaFold3 [2] would be promising directions. Second, the integration of pretrained protein language models [55] and structure models [96] could significantly enhance PocketFlow's performance. Additionally, wet lab experiments to verify PocketFlow's efficacy are planned. Potential negative impacts may include the misuse of PocketFlow for creating harmful biomolecules [32]. Rigorous oversight and screening access to the model should be considered.

## 6 Conclusion

In this paper, we proposed PocketFlow, a protein-ligand interaction prior-informed flow matching model for protein pocket generation. We define multimodal flow matching for protein backbone

frames, sidechain torsion angles, and residue/interaction types to appropriately represent the protein-ligand complex. The binding affinity and interaction geometry guidance effectively improve the validity and affinity of the generated pockets. Moreover, PocketFlow offers a unified framework covering small-molecule, nucleic acids, and peptides-binding protein pocket generation.

## 7   Acknowledgements

This research was supported by grants from the National Natural Science Foundation of China (Grant No. 623B2095) and the Fundamental Research Funds for the Central Universities.

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

# A Dataset Preprocessing

We consider two widely used datasets for benchmark evaluation: **CrossDocked** dataset [27] contains 22.5 million protein-molecule pairs generated through cross-docking. Following previous works [59, 64, 92], we filter out data points with binding pose RMSD greater than 1 Å, leading to a refined subset with around 180k data points. For data splitting, we use mmseqs2 [75] to cluster data at 30% sequence identity, and randomly draw 100k protein-ligand structure pairs for training and 100 pairs from the remaining clusters for testing and validation, respectively; **Binding MOAD** dataset [34] contains around 41k experimentally determined protein-ligand complexes. Following previous work [71], we keep pockets with valid and moderately 'drug-like' ligands with QED score $\geq 0.3$. We further filter the dataset to discard molecules containing atom types $\notin \{C, N, O, S, B, Br, Cl, P, I, F\}$ as well as binding pockets with non-standard amino acids. Then, we randomly sample and split the filtered dataset based on the Enzyme Commission Number (EC Number) [9] to ensure different sets do not contain proteins from the same EC Number main class. Finally, we have 40k protein-ligand pairs for training, 100 pairs for validation, and 100 pairs for testing. For all the benchmark tasks in this paper, PocketFlow and all the other baseline methods are trained with the same data split for a fair comparison.

To test the generalizability of PocketFlow to other ligand modalities, we further consider **PPDBench** [3], which contains 133 non-redundant complexes of protein-peptides and **PDBBind RNA** [80], which contains 56 protein-RNA pairs by filtering the PDBBind nucleic acid subset with RNA sequence lengths longer than 5 and less than 15.

# B Considered Protein-ligand Interactions

Table 4: Key geometric constraints to define protein-ligand interactions [69]. Angles in degree and distances in Ångström.

| Variable | Value | Description | Ref. |
|---|---|---|---|
| INTER_DIST_MIN | 2.0 Å | Min. distance to avoid steric clashes | [31] |
| HYDROPH_DIST_MAX | 4.0 Å | Max. distance of carbon atoms for a hydrophobic interaction | [69] |
| HBOND_DIST_MAX | 4.1 Å | Max. distance between acceptor and donor in hydrogens bonds | [35] |
| HBOND_DON_ANGLE_MIN | 100° | Min. angle at the hydrogen bond donor (X-D...A) | [35] |
| HBOND_ACC_ANGLE_MIN | 100° | Min. angle at the hydrogen bond acceptor (X-A...D) | [35] |
| PISTACK_DIST_MAX | 7.5 Å | Max. distance between ring centers for stacking | [26] |
| PISTACK_ANG_DEV | 30° | Max. deviation from optimum angle for stacking | [69] |
| PISTACK_OFFSET_MAX | 2.0 Å | Max. offset between aromatic ring centers for stacking | [69] |
| SALTBRIDGE_DIST_MAX | 5.5 Å | Distance between two centers of charges in salt bridges | [10] |

Following [97], we considered 4 dominant non-covalent interaction types in PocketFlow, including salt bridges, $\pi-\pi$ stacking, hydrogen bonds, and hydrophobic interactions (ranked based on their contribution to affinity and reversed frequency). The frequency statistics are listed in Table.3.

- **Salt bridges** [10], which are electrostatic interactions between oppositely charged centers, are often considered among the strongest interactions in protein structures and other biomolecular complexes. They can significantly contribute to stability and binding affinity due to their strong electrostatic nature. To form salt bridges, two centers of opposite charges need to be below a distance of SALTBRIDGE_DIST_MAX.

- **Hydrogen bonds** [35] occur between a hydrogen atom covalently bonded to a more electronegative atom (like oxygen or nitrogen) and another electronegative atom. Their strength is less than that of salt bridges but is significant in biological contexts.

  A hydrogen bond is established between a hydrogen bond donor and acceptor (OpenBabel [63] is used to detect hydrogen bond donor/acceptor). The distance between the donor and acceptor needs to be less than HBOND_DIST_MAX. The donor and acceptor angle needs to be larger than HBOND_DON_ANGLE_MIN and HBOND_ACC_ANGLE_MIN respectively. Since PocketFlow only considers heavy atoms (no hydrogen atoms), we consider the geometry

of hydrogen bonds without protonation [35] (see Figure. 4). For simplicity, we do not differentiate donor/acceptor in the interaction geometry guidance.

- $\pi$–$\pi$ **stacking** [69] involve the stacking of aromatic rings (like those found in phenylalanine, tyrosine, or tryptophan) due to favorable van der Waals forces and sometimes electrostatic interactions. $\pi$–$\pi$ stacking is crucial in the structure of nucleic acids and proteins, especially in the active sites of many enzymes, although they are generally weaker than hydrogen bonds and salt bridges.

  To form $\pi$–$\pi$ stacking, we first need two aromatic rings (OpenBabel [63] is used to detect aromatic rings). The distance between the two ring centers needs to be below `PISTACK_DIST_MAX`. The angle between two normal vectors of ring planes needs to be below `PISTACK_ANG_DEV`. Additionally, each ring center is projected onto the opposite ring plane. The distance between the other ring center and the projected point (i.e., the offset) has to be less than `PISTACK_OFFSET_MAX`. Figure. 5 shows the illustration.

- **Hydrophobic Interactions** [11] are caused by the tendency of hydrophobic side chains to avoid contact with water, leading them to aggregate. While these are not strong interactions on their own, they play a crucial role in the folding and stability of proteins by driving the burial of nonpolar groups away from the aqueous environment, thereby contributing significantly to the overall stability. To form hydrophobic interactions, the atom distance needs to be less than `HYDROPH_DIST_MAX`.

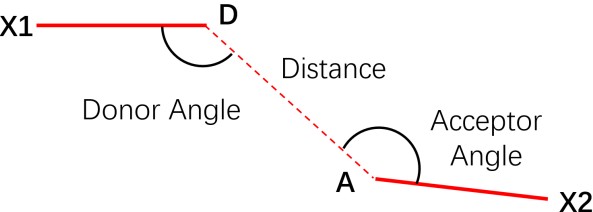

Figure 4: Schematic representation of the geometry of a **hydrogen bond** (without protonation). D and A denote the hydrogen bond donor and acceptor respectively. X1 and X2 are the neighboring atoms of donor and acceptor. The hydrogen bond distance as well as donor/acceptor angles are illustrated.

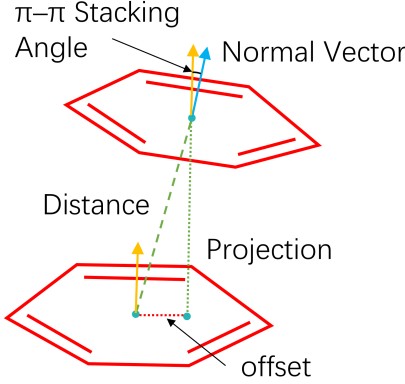

Figure 5: Schematic representation of the geometry of $\pi$–$\pi$ **stacking**. To form a $\pi$–$\pi$ stacking, we need two aromatic rings. The distance of ring centers, the angle between normal vectors, and the projection offset of the ring centers need to satisfy a set of geometry constraints.

## C   Model Details

In PocketFlow, we adopt a neural network architecture modified from the FrameDiff [86]. This architecture consists of Invariant Point Attention from AlphaFold2 [43] and transformer blocks

[79]. In this section, we use superscripts to refer to the network layer and subscripts to indexes or variables. In the network, each residue/ligand atom is represented by one embedding $h \in \mathbb{R}^{D_x}$ and a frame $T \in SE(3)$. For the ligand atoms, the orientation matrix of frame is set as identity matrices. Overall, at the $\ell$-th layer of the network, $\boldsymbol{h}^\ell = [h_1^\ell, \ldots, h_N^\ell] \in \mathbb{R}^{N \times D_x}$ are all the node embeddings where $h_i^\ell$ is the embedding for the $i$-th node and $N = N_p + N_l$ is the total number of nodes; $\boldsymbol{T}^\ell = [T_1^\ell, \ldots, T_N^\ell] \in SE(3)^N$ is the frames of every node at the $\ell$-th layer; $\boldsymbol{z}^\ell \in \mathbb{R}^{N \times N \times D_z}$ are edge embeddings with $z_{ij}^\ell$ being the embedding of the edge between residues $i$ and $j$. In the following paragraphs, we introduce the details of feature initialization, node/edge update, backbone update, and residue type/interaction type/sidechain torsion angle predictions.

**Feature initialization.** Following [86], node embeddings are initialized with residue indices and timestep while edge embeddings additionally get relative sequence distances. Initial embeddings at layer 0 for residues $i, j$ are obtained with an MLP and sinusoidal embeddings $\phi(\cdot)$ [79] over the features. Following [86], we additionally include self-conditioning of predicted $C_\alpha$ displacements. Let $\tilde{x}$ be the $C_\alpha$ coordinates predicted during self-conditioning. 50% of the time we set $\tilde{x} = 0$. The binned displacement of two $C_\alpha$ is given as,

$$\text{disp}_{ij} = \sum_{k=1}^{N_{\text{bins}}} 1\{\tilde{x}^i - \tilde{x}^j < d_k\}, \tag{14}$$

where $d_1, \ldots, d_{N_{\text{bins}}}$ are linspace(0, 20) are equally spaced bins between 0 and 20 angstroms. In our experiments we set $N_{\text{bins}} = 22$. The initial embeddings can be expressed as

$$h_i^0 = \text{MLP}(\phi(i), \phi(t)), \quad h_i^0 \in \mathbb{R}^{D_h}, \tag{15}$$

$$z_{ij}^0 = \text{MLP}(\phi(i), \phi(j), \phi(j-i), \phi(t), \phi(\text{disp}_{ij})), \quad z_{ij}^0 \in \mathbb{R}^{D_z}, \tag{16}$$

where $D_h, D_z$ are node and edge embedding dimensions. For the initialization of $C_\alpha$ coordinates, we use the interpolation and extrapolation strategy of FAIR [92].

**Node update.** The process of node update is shown below. Invariant Point Attention (IPA) is from [43]. No weight sharing is performed across layers. We use the vanilla Transformer from [79]. We use Multi-Layer Perceptrons (MLP) with 3 Linear layers, ReLU activation, and LayerNorm [6] after the final layer.

$$\boldsymbol{h}_{\text{ipa}} = \text{LayerNorm}(\text{IPA}(\boldsymbol{h}^\ell, \boldsymbol{z}^\ell, \boldsymbol{T}^\ell) + \boldsymbol{h}^\ell), \quad \boldsymbol{h}_{\text{ipa}} \in \mathbb{R}^{N \times D_h} \tag{17}$$

$$\boldsymbol{h}_{\text{skip}} = \text{Linear}(\boldsymbol{h}_0), \quad \boldsymbol{h}_{\text{skip}} \in \mathbb{R}^{N \times D_{\text{skip}}} \tag{18}$$

$$\boldsymbol{h}_{\text{in}} = \text{concat}(\boldsymbol{h}_{\text{ipa}}, h_{\text{skip}}), \quad \boldsymbol{h}_{\text{in}} \in \mathbb{R}^{N \times (D_h + D_{\text{skip}})} \tag{19}$$

$$\boldsymbol{h}_{\text{trans}} = \text{Transformer}(\boldsymbol{h}_{\text{in}}), \quad \boldsymbol{h}_{\text{trans}} \in \mathbb{R}^{N \times (D_h + D_{\text{skip}})} \tag{20}$$

$$\boldsymbol{h}_{\text{out}} = \text{Linear}(\boldsymbol{h}_{\text{trans}}) + \boldsymbol{h}_\ell, \quad \boldsymbol{h}_{\text{out}} \in \mathbb{R}^{N \times D_h} \tag{21}$$

$$\boldsymbol{h}^{\ell+1} = \text{MLP}(\boldsymbol{h}_{\text{out}}), \quad \boldsymbol{h}_{\ell+1} \in \mathbb{R}^{N \times D_h} \tag{22}$$

**Edge update.** Each edge is updated with a MLP over the current edge and source and target node embeddings. In the first line, node embeddings are first projected down to half the dimension.

$$\boldsymbol{h}_{\text{down}}^{\ell+1} = \text{Linear}(\boldsymbol{h}^{\ell+1}), \quad \boldsymbol{h}_{\text{down}}^{\ell+1} \in \mathbb{R}^{N \times D_h/2} \tag{23}$$

$$\boldsymbol{z}_{ij}' = \text{concat}(\boldsymbol{h}_{\text{down},i}^{\ell+1}, \boldsymbol{h}_{\text{down},j}^{\ell+1}, \boldsymbol{z}_{ij}^\ell), \quad \boldsymbol{z}_{ij}' \in \mathbb{R}^{N \times (2D_h + D_z)} \tag{24}$$

$$\boldsymbol{z}^{\ell+1} = \text{LayerNorm}(\text{MLP}(\boldsymbol{z}')), \quad \boldsymbol{z}^{\ell+1} \in \mathbb{R}^{N \times N \times D_z} \tag{25}$$

**Backbone update.** Our frame updates follow the BackboneUpdate algorithm in AlphaFold2 [43]. We write the algorithm here with our notation,

$$(b_i, c_i, d_i, x_i^{\text{update}}) = \text{Linear}(h_i^L), \tag{26}$$

$$(a_i, b_i, c_i, d_i) = (1, b_i, c_i, d_i) / \sqrt{1 + b_i^2 + c_i^2 + d_i^2}, \tag{27}$$

$$R_i^{\text{update}} = \begin{pmatrix} a_i^2 + b_i^2 - c_i^2 - d_i^2 & 2b_ic_i - 2a_id_i & 2b_id_i + 2a_ic_i \\ 2b_ic_i + 2a_id_i & a_i^2 - b_i^2 + c_i^2 - d_i^2 & 2c_id_i - 2a_ib_i \\ 2b_id_i - 2a_ic_i & 2c_id_i + 2a_ib_i & a_i^2 - b_i^2 - c_i^2 + d_i^2 \end{pmatrix}, \tag{28}$$

$$T_i^{\text{update}} = (R_i^{\text{update}}, x_i^{\text{update}}), \tag{29}$$

$$T_i^{\ell+1} = T_i^{\ell} \cdot T_i^{\text{update}}, \tag{30}$$

where $b_i, c_i, d_i \in \mathbb{R}, x_i^{\text{update}} \in \mathbb{R}^3$. Equ. 27 constructs a normalized quaternion which is then converted into a valid rotation matrix in Equ. 28. Following [84, 81], we use the planar geometry of the backbone to impute the oxygen atoms. Note that we only update the pocket and ligand nodes in PocketFlow while setting the scaffold nodes fixed.

**Residue/Interaction Type and Torsion angle Prediction.** We predict the residue/interaction types and sidechain torsion angles based on node embeddings.

$$\boldsymbol{h_c} = \text{MLP}(\boldsymbol{h}^L), \quad \boldsymbol{h}_I = \text{MLP}(\boldsymbol{h}^L), \quad \boldsymbol{h_\chi} = \text{MLP}(\boldsymbol{h}^L), \tag{31}$$

$$\boldsymbol{c} = \text{softmax}(\text{Linear}(\boldsymbol{h_c} + \boldsymbol{h}^L)), I = \text{softmax}(\text{Linear}(\boldsymbol{h_\psi} + \boldsymbol{h}^L)), \tag{32}$$

$$\boldsymbol{\chi} = \text{Linear}(\boldsymbol{h_\chi} + \boldsymbol{h}^L)\text{mod}2\pi \tag{33}$$

where $\boldsymbol{c} \in \mathbb{R}^{N \times 20}, I \in \mathbb{R}^{N,5}$, and $\boldsymbol{\chi} \in [0, 2\pi)^{4N}$. In PocketFlow, the number of network blocks is set to 8, the number of transformer layers within each block is set to 4, and the number of hidden channels used in the IPA calculation is set to 16. The node embedding size $D_h$ and the edge embedding size $D_z$ are set as 128. We removed skip connections and psi-angle prediction. For model training, we use Adam [45] optimizer with learning rate 0.0001, $\beta_1 = 0.9$, $\beta_2 = 0.999$. We train on a Tesla A100 GPU for 20 epochs. In the sampling process, the total number of steps $T$ is set as 50.

# D  Proof of Equivariance

**Theorem 1.** *Denote the SE(3)-transformation as $T_g$, PocketFLow $p_\theta(\mathcal{R}, \mathcal{G}|\mathcal{P} \setminus \mathcal{R})$ is SE(3) equivariant i.e., $p_\theta(T_g(\mathcal{R}, \mathcal{G})|T_g(\mathcal{P} \setminus \mathcal{R})) = p_\theta(\mathcal{R}, \mathcal{G}|\mathcal{P} \setminus \mathcal{R})$, where $\mathcal{R}$ denotes the designed pocket, $\mathcal{G}$ is the binding ligand, and $\mathcal{P} \setminus \mathcal{R}$ is the protein scaffold.*

*Proof.* The main idea is that the SE(3)-invariant prior and SE(3)-equivariant neural network lead to an SE(3)-equivariant generative process of PocketFlow. By subtracting the CoM of the scaffold from the initialized structure, we obtain an SE(3)-invariant prior distribution similar to [86, 29]. Moreover, the neural network for structure update as shown in Appendix C is SE(3)-equivariant. Formally, the two conditions to guarantee an invariant likelihood $p_\theta(\mathcal{R}_1, \mathcal{G}_1|\mathcal{P} \setminus \mathcal{R})$ are as follows (we use subscripts to denote the time steps from $t = 0$ to $t = 1$):

$$\text{Invariant Prior:} \quad p(\mathcal{R}_0, \mathcal{G}_0, \mathcal{P} \setminus \mathcal{R}) = p(T_g(\mathcal{R}_0, \mathcal{G}_0, \mathcal{P} \setminus \mathcal{R})), \tag{34}$$

$$\text{Equivariant Transition:} \quad p_\theta(\mathcal{R}_{t+\Delta t}, \mathcal{G}_{t+\Delta t}|\mathcal{R}_t, \mathcal{G}_t, \mathcal{P}\setminus\mathcal{R}) = p_\theta(T_g(\mathcal{R}_{t+\Delta t}, \mathcal{G}_{t+\Delta t})|T_g(\mathcal{R}_t, \mathcal{G}_t, \mathcal{P}\setminus\mathcal{R})), \tag{35}$$

We can obtain the conclusion as follows:

$$p_\theta(T_g(\mathcal{R}_1, \mathcal{G}_1)|T_g(\mathcal{P} \setminus \mathcal{R})) = \int p(T_g(\mathcal{R}_0, \mathcal{G}_0, \mathcal{P} \setminus \mathcal{R})) \prod_{s=0}^{T-1} p_\theta(T_g(\mathcal{R}_{(s+1)\Delta t}, \mathcal{G}_{(s+1)\Delta t})|T_g(\mathcal{R}_{s\Delta t}, \mathcal{G}_{s\Delta t}, \mathcal{P} \setminus \mathcal{R}))$$

$$= \int p(\mathcal{R}_0, \mathcal{G}_0, \mathcal{P} \setminus \mathcal{R}) \prod_{s=0}^{T-1} p_\theta(T_g(\mathcal{R}_{(s+1)\Delta t}, \mathcal{G}_{(s+1)\Delta t})|T_g(\mathcal{R}_{s\Delta t}, \mathcal{G}_{s\Delta t}, \mathcal{P} \setminus \mathcal{R}))$$

$$= \int p(\mathcal{R}_0, \mathcal{G}_0, \mathcal{P} \setminus \mathcal{R}) \prod_{s=0}^{T-1} p_\theta(\mathcal{R}_{(s+1)\Delta t}, \mathcal{G}_{(s+1)\Delta t}|\mathcal{R}_{s\Delta t}, \mathcal{G}_{s\Delta t}, \mathcal{P} \setminus \mathcal{R})$$

$$= p_\theta(\mathcal{R}_1, \mathcal{G}_1|\mathcal{P} \setminus \mathcal{R}),$$

where $T$ is the total number of steps. We apply the invariant prior and equivariant transition conditions in the derivation. $\square$

# E Classifier-guided Flow Matching

Here, we present the Bayesian approach to guide the flow matching with the affinity predictor. The key insight comes from connecting flow matching to diffusion models to which affinity guidance can be applied. Sampling a data point from the prior distribution $p_0$, we have the following ordinary differential equation (ODE) [72] that pushes it to data distribution:

$$dC_t = v(C_t, t)dt = \left[ f(C_t, t) - \frac{1}{2} g(t)^2 \nabla \log p_t(C_t) \right] dt, \tag{36}$$

where $\nabla \log p_t(C_t)$ is the score function, $f(C_t, t)$ and $g(t)$ are the drift and diffusion coefficients respectively. We modify Equ. 36 to be conditioned on the affinity label ($v_b = 1$) followed by an application of Bayes rule,

$$dC_t = \left[ f(C_t, t) - \frac{1}{2} g(t)^2 \nabla \log p_t(C_t | v_b = 1) \right] dt \tag{37}$$

$$= \left[ f(C_t, t) - \frac{1}{2} g(t)^2 \left( \nabla \log p_t(C_t) + \nabla \log p_t(v_b = 1 | C_t) \right) \right] dt \tag{38}$$

$$= \left[ v(C_t, t) - \frac{1}{2} g(t)^2 \nabla \log p_t(v_b = 1 | C_t) \right] dt, \tag{39}$$

where the first term is the unconditional vector field and the second term is the affinity guidance term. In practice, we do not directly predict the affinity label based on $C_t$ because the intermediate structure is noisy. We use the following transformation:

$$p_t(v_b = 1 | C_t) = \int p(v_b = 1 | C_1) p(C_1 | C_t) dC_t \approx p(v_b = 1 | \hat{C}_1(C_t)), \tag{40}$$

where $\hat{C}_1(C_t)$ is the expected denoised protein-ligand complex structure based on $C_t$. Details of the affinity predictor is introduced in the Appendix. E.1. We need to choose $g(t)$ such that it matches the learned probability path. Previous works [20, 86] showed $g(t)^2 = \frac{t}{1-t}$ in the Euclidean setting. For simplicity, we set $g(t)$ as constant 1 and observe good performance in experiments.

## E.1 Binding Affinity Predictor

In PocketFlow, we leverage a binding affinity predictor $p_\theta(v_b | \hat{C}_1(C_t))$ to guide the denoising process, where $v_b \in \{0, 1\}$ is the binary label of binding affinity and $\hat{C}_1(C_t)$ is the expected protein-ligand structure at $t = 1$. Following [66, 29], we leverage a 3-layer EGNN [70] with the node initialized embeddings and residue/ligand atom coordinates from Appendix C. Specifically, we take the $C_\alpha$ coordinates for the residues and ligand atom coordinates and construct $k$-NN graphs ($k$ set as 9). Let $h_i^\ell$ and $x_i^\ell$ be denote the node representations and coordinates at the $\ell-$th layer. The $(\ell + 1)-$th layer is computed as follows:

$$h_i^{\ell+1} = h_i^\ell + \sum_{j \in \mathcal{N}, i \neq j} f_h(d_{ij}^\ell, h_i^\ell, h_j^\ell, e_{ij}), \tag{41}$$

$$x_i^{\ell+1} = x_i^\ell + \sum_{j \in \mathcal{N}, i \neq j} (x_i^\ell - x_j^\ell) f_x(d_{ij}^\ell, x_i^\ell, x_j^\ell, e_{ij}), \tag{42}$$

where $d_{ij}^\ell = \|x_i^\ell - x_j^\ell\|$ represents the Euclidean distance between node $i$ and node $j$ at the $\ell$-th layer, $\mathcal{N}$ denotes the $k$-NN neighbors, and $e_{ij}$ indicates the direction of message-passing, including from protein to protein, from protein to ligand, from ligand to protein, and from ligand to ligand. The functions $f_h$ and $f_x$ are graph attention networks. Finally, we append an average pooling, one linear layer, and softmax operation at the end to predict the binary label of affinity.

To train the binding affinity predictor, we first annotate the data points in the corresponding training set: data points are annotated 1 if their affinity is higher than the average score of the dataset, otherwise 0. We train the predictor separately instead of joint training with flow matching because we find it can converge more quickly than the flow matching losses. We did not train the predictor on the intermediate structures as we find they are noisy and deteriorate the predictor and PocketFlow's overall performance. In experiments, we use the Adam optimizer and train for 10 epochs.

## E.2 Geometry Guidance

**Distance Guidance.** For hydrogen bonds, the distances between donor and acceptor atoms need to be less than 4.1 Å and larger than 2 Å to reduce steric clashes [35]. The following inequality is a necessary condition for residues in $\hat{\mathcal{C}}_1(\mathcal{C}_t)$ with predicted interaction label $\hat{I}_1$ as hydrogen bond:

$$l_{\min} \leq \min_{i \in \mathcal{A}_{hbond}^{(k)}, j \in \mathcal{G}} \left\| \boldsymbol{x}^{(i)} - \boldsymbol{x}^{(j)} \right\|_2 \leq l_{\max}, \tag{43}$$

where $l_{\min}$ and $l_{\max}$ are distance constraints; $\mathcal{A}_{hbond}^{(k)}$ denote the $k$-th residue in the set of pocket residues with predicted hydrogen bonds. With a little abuse of notations, $\boldsymbol{x}^{(i)}$ and $\boldsymbol{x}^{(j)}$ denote the atom coordinates in the residue and ligand respectively. We use the following derivations to obtain the guidance term for the distance constraints:

$$\nabla_{\mathcal{C}_t} \log P(\{l_{\min} \leq \min_{i \in \mathcal{A}_{hbond}^{(k)}, j \in \mathcal{G}} \|\boldsymbol{x}^{(i)} - \boldsymbol{x}^{(j)}\|_2 \leq l_{\max}, k = 1 : |\mathcal{A}_{hbond}|\}) \tag{44}$$

$$= \nabla_{\mathcal{C}_t} \sum_{k=1}^{|\mathcal{A}_{hbond}|} \log P(l_{\min} \leq \min_{i \in \mathcal{A}_{hbond}^{(k)}, j \in \mathcal{G}} \|\boldsymbol{x}^{(i)} - \boldsymbol{x}^{(j)}\|_2 \leq l_{\max}) \tag{45}$$

$$= \sum_{k=1}^{|\mathcal{A}_{hbond}|} \frac{\nabla_{\mathcal{C}_t}[P(-\min_{i \in \mathcal{A}_{hbond}^{(k)}, j \in \mathcal{G}} \|\boldsymbol{x}^{(i)} - \boldsymbol{x}^{(j)}\|_2 \leq -l_{\min}) \cdot P(\min_{i \in \mathcal{A}_{hbond}^{(k)}, j \in \mathcal{G}} \|\boldsymbol{x}^{(i)} - \boldsymbol{x}^{(j)}\|_2 \leq l_{\max})]}{P(l_{\min} \leq \min_{i \in \mathcal{A}_{hbond}^{(k)}, j \in \mathcal{G}} \|\boldsymbol{x}^{(i)} - \boldsymbol{x}^{(j)}\| \leq l_{\max})} \tag{46}$$

$$= \sum_{k=1}^{|\mathcal{A}_{hbond}|} \xi_1 \nabla_{\mathcal{C}_t} P(\min_{i \in \mathcal{A}_{hbond}^{(k)}, j \in \mathcal{G}} \|\boldsymbol{x}^{(i)} - \boldsymbol{x}^{(j)}\|_2 \leq l_{\max}) + \xi_2 \nabla_{x_t} P(-\min_{i \in \mathcal{A}_{hbond}^{(k)}, j \in \mathcal{G}} \|\boldsymbol{x}^{(i)} - \boldsymbol{x}^{(j)}\|_2 \leq -l_{\min}) \tag{47}$$

$$= \sum_{k=1}^{|\mathcal{A}_{hbond}|} \xi_1 \nabla_{\mathcal{C}_t} \mathbb{I}(\min_{i \in \mathcal{A}_{hbond}^{(k)}, j \in \mathcal{G}} \|\boldsymbol{x}^{(i)} - \boldsymbol{x}^{(j)}\|_2 \leq l_{\max}) + \xi_2 \nabla_{\mathcal{C}_t} \mathbb{I}(-\min_{i \in \mathcal{A}_{hbond}^{(k)}, j \in \mathcal{G}} \|\boldsymbol{x}^{(i)} - \boldsymbol{x}^{(j)}\|_2 \leq -l_{\min}), \tag{48}$$

where $\xi_1 = 1/\nabla_{\mathcal{C}_t} P(\min_{i \in \mathcal{A}_{hbond}^{(k)}, j \in \mathcal{G}} \|\boldsymbol{x}^{(i)} - \boldsymbol{x}^{(j)}\|_2 \leq l_{\max})$ and $\xi_2 = 1/P(-\min_{i \in \mathcal{A}_{hbond}^{(k)}, j \in \mathcal{G}} \|\boldsymbol{x}^{(i)} - \boldsymbol{x}^{(j)}\|_2 \leq -l_{\min})$. Due to the discontinuity of the indicator function $\mathbb{I}(\cdot)$ that is incompatible with the gradient, we apply $\xi - \max(0, \xi - y)$ as a surrogate of $\mathbb{I}(y < \xi)$ in the above equation. Although $\xi_1$ and $\xi_2$ are dependent on $\mathcal{C}_t$, we find setting them as constant still works well in experiments. With these approximations, we can derive guidance term for hydrogen bond distance constraints:

$$-\nabla_{\mathcal{C}_t} \sum_{k=1}^{|\mathcal{A}_{hbond}|} \left[ \xi_1 \max\left(0, d^{(k)} - l_{\max}\right) + \xi_2 \max\left(0, l_{\min} - d^{(k)}\right) \right], \tag{49}$$

where $d^{(k)} = \min_{i \in \mathcal{A}_{hbond}^{(k)}, j \in \mathcal{G}} \left\| \boldsymbol{x}^{(i)} - \boldsymbol{x}^{(j)} \right\|_2$. Such distance guidance terms for hydrophobic interactions, salt bridges, and $\pi - \pi$ stackings are similar. The difference is to replace $\mathcal{A}_{hbond}$ with $\mathcal{A}_{hydro}$, $\mathcal{A}_{salt}$, and $\mathcal{A}_\pi$ that denotes the residue sets with corresponding interactions. We modify the functions in plip [2] for the ease of detecting interaction atom pair candidates. In practice, $\nabla_{\mathcal{C}_t}$ takes gradients with each component in $\mathcal{C}_t$, including $\boldsymbol{\chi}_t, \boldsymbol{x}_t, \boldsymbol{O}_t, \boldsymbol{c}_t$, and $I_t$.

**Angle Guidance.** Besides the distance constraint, the hydrogen bond needs to satisfy the acceptor/donor angle constraint [69], e.g., the donor/acceptor angle needs to be larger than $100°$. hangle$(\cdot, \cdot)$

---

[2]https://github.com/pharmai/plip

calculates the acceptor/donor angle in Figure. 4.

$$\nabla_{\mathcal{C}_t} \log P(\{\alpha_{\min} \leq \max_{i \in \mathcal{A}_{hbond}^{(k)}, j \in \mathcal{G}} \text{hangle}(\boldsymbol{x}^{(i)}, \boldsymbol{x}^{(j)}), k = 1 : |\mathcal{A}_{hbond}|\}) \tag{50}$$

$$= \nabla_{\mathcal{C}_t} \sum_{k=1}^{|\mathcal{A}_{hbond}|} \log P(\alpha_{\min} \leq \max_{i \in \mathcal{A}_{hbond}^{(k)}, j \in \mathcal{G}} \text{hangle}(\boldsymbol{x}^{(i)}, \boldsymbol{x}^{(j)})) \tag{51}$$

$$= \sum_{k=1}^{|\mathcal{A}_{hbond}|} \frac{\nabla_{\mathcal{C}_t} P(\alpha_{\min} \leq \max_{i \in \mathcal{A}_{hbond}^{(k)}, j \in \mathcal{G}} \text{hangle}(\boldsymbol{x}^{(i)}, \boldsymbol{x}^{(j)}))}{P(\alpha_{\min} \leq \max_{i \in \mathcal{A}_{hbond}^{(k)}, j \in \mathcal{G}} \text{hangle}(\boldsymbol{x}^{(i)}, \boldsymbol{x}^{(j)}))} \tag{52}$$

$$= \sum_{k=1}^{|\mathcal{A}_{hbond}|} \xi_3 \nabla_{\mathcal{C}_t} P(\alpha_{\min} \leq \max_{i \in \mathcal{A}_{hbond}^{(k)}, j \in \mathcal{G}} \text{hangle}(\boldsymbol{x}^{(i)}, \boldsymbol{x}^{(j)})), \tag{53}$$

where $\xi_3 = 1/P(\alpha_{\min} \leq \max_{i \in \mathcal{A}_{hbond}^{(k)}, j \in \mathcal{G}} \text{hangle}(\boldsymbol{x}^{(i)}, \boldsymbol{x}^{(j)}))$. The final guidance term is:

$$-\xi_3 \nabla_{x_t} \sum_{k=1}^{|\mathcal{A}_{hbond}|} \max(0, \alpha_{\min} - \phi^{(k)}), \tag{54}$$

where $\phi^{(k)} = \max_{i \in \mathcal{A}_{hbond}^{(k)}, j \in \mathcal{G}} \text{hangle}(\boldsymbol{x}^{(i)}, \boldsymbol{x}^{(j)})$. The angle constraint is similar for the $\pi - \pi$ stacking and the final guidance term is:

$$-\xi_4 \nabla_{\mathcal{C}_t} \sum_{k=1}^{|\mathcal{A}_\pi|} \max(0, \phi_\pi^{(k)} - \alpha_{\max}), \tag{55}$$

where $\phi_\pi^{(k)} = \min_{i \in \mathcal{A}_\pi^{(k)}, j \in \mathcal{G}} \text{piangle}(\boldsymbol{x}^{(i)}, \boldsymbol{x}^{(j)})$ and $\text{piangle}(\cdot, \cdot)$ calculates the $\pi - \pi$ stacking angle in Figure. 4. All the operations and calculations used in geometry guidance are made differentiable and can be plugged into the sampling process of PocketFlow.

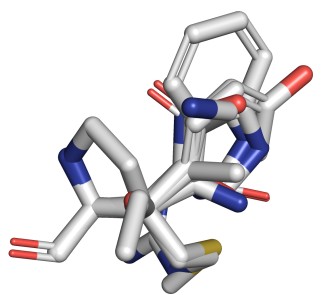

Figure 6: Superposition of 20 residue-type sidechains. When calculating the geometry guidance, we use the expected sidechain conformations with respect to the estimated residue type probability $\hat{\boldsymbol{c}}_1^{(i)}$ to avoid the non-differentiability issue of residue type sampling.

**Sidechain Ensemble.** PocketFlow takes the co-design scheme, where the residue type/side chain structure of the pocket is not determined during sampling. Directly sampling from the residue type distribution makes the model not differentiable [38]. We propose to use the sidechain ensemble for the interaction geometry calculation, i.e., the weighted sum of geometric guidance with respect to residue types. For example, for Equ. 54, we have:

$$-\xi_3 \nabla_{\mathcal{C}_t} \sum_{k=1}^{|\mathcal{A}_{hbond}|} \sum_{n=1}^{20} \hat{\boldsymbol{c}}_1^{(i)}[n] \cdot \max(0, \alpha_{\min} - \phi^{(k)}), \tag{56}$$

where $\hat{\boldsymbol{c}}_1^{(i)}[n]$ denote the $n$-th residue type probability and $\phi^{(k)}$ calculates the angle with the $n$-th type residue side chain.

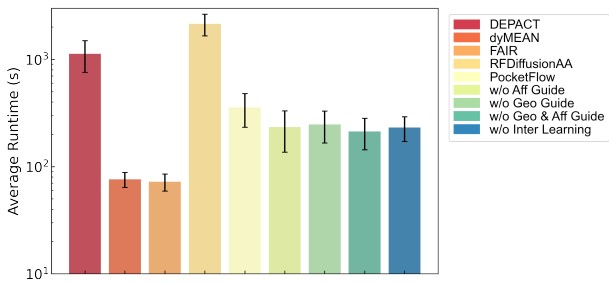

Figure 7: Average Generation time for 100 pockets by different models on CrossDocked (the error bars show the standard deviations over different runs).

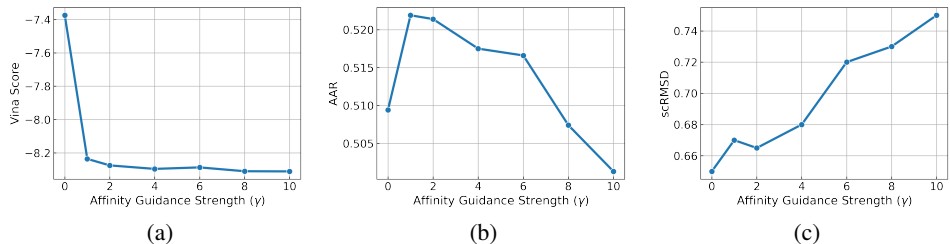

Figure 8: The influence of Affinity Guidance Strength $\gamma$ on the pocket metrics.

# F    More Results

Here, we show additional results on efficiency analysis (Figure. 7), hyperparameter analysis of $\gamma$ (Figure. 8), and ablation studies on the interaction analysis (Table. 6).

Figure. 7 shows that the PocketFlow is much more efficient than stat-of-the-art diffusion-based models such as RFDiffusionAA. Considering the high quality of the generated pockets, the slight time overhead over models based on iterative refinement (e.g., dyMEAN and FAIR) is acceptable. We find that Affinity and Interaction Geometry Guidance do not add much overhead to the generation process. Therefore, these prior guidance are efficient tools for pocket optimization.

In Figure. 8, we explore the impact of Affinity Guidance Strength ($\gamma$) on various generation metrics. As $\gamma$ is scaled up, the Vina Score significantly improves and quickly stabilizes; AAR initially increases before gradually decreasing; scRMSD, on the other hand, increases with higher $\gamma$. These observations underscore the importance of selecting an appropriate $\gamma$ to effectively balance the guidance and unconditional terms. While Affinity Guidance promotes the generation of high-affinity pockets, an excessively high $\gamma$ can result in less valid pocket sequences or structures. In the default configuration, $\gamma$ is set to 1.

To evaluate the validity of the generated sidechain structure, we compute the Mean Absolute Error (MAE) of sidechain angles (degrees) following [90] in Table. 5. We mainly compare PocketFlow with RFDiffusionAA [48]+LigandMPNN [22] on the recovered residues. In the table, we report the average MAE and can observe that PocketFlow achieves better performance in generating valid sidechain structures.

| Method | $\chi_1$ | $\chi_2$ | $\chi_3$ | $\chi_4$ |
|---|---|---|---|---|
| RFDiffusionAA | 21.56 | 27.92 | 48.76 | 52.88 |
| PocketFlow | **19.40** | **26.22** | **44.57** | **50.10** |

Table 5: The MAE of RFDiffusionAA + LigandMPNN and PocketFlow on sidechain torsion angles(degrees).

In Table. 6, we supplement further results of interaction analysis (Table. 3 in the main paper). We can observe that the guidance terms effectively improve the number of favorable interactions while reducing steric clashes, which lay the foundation for generating high-affinity pockets.

| Methods | Clash ($\downarrow$) | HB ($\uparrow$) | Salt ($\uparrow$) | Hydro ($\uparrow$) | $\pi$–$\pi$ ($\uparrow$) |
|---|---|---|---|---|---|
| PocketFlow | **1.21** | **4.12** | **0.27** | **6.03** | **0.28** |
| w/o Aff Guide | 2.58 | 3.84 | 0.25 | 5.84 | 0.27 |
| w/o Geo Guide | 3.27 | 3.96 | 0.24 | 5.90 | 0.27 |
| w/o Geo & Aff Guide | 3.56 | 3.68 | 0.23 | 5.73 | 0.26 |
| w/o Inter Learning | 3.34 | 3.74 | 0.22 | 5.80 | 0.26 |

Table 6: Ablation studies on the interaction analysis. The best results are bolded and the runner-up is underlined.

## G   Baseline Implementation

**DEPACT** [19] [3] is a template-matching method that follows a two-step strategy for pocket design. Firstly, it searches the protein-ligand complexes in the template database with similar ligand fragments and constructs a cluster model (a set of pocket residues). The template databases are constructed based on the corresponding training datasets for fair comparisons. Secondly, it grafts the cluster model into the protein pocket with PACMatch. It works by placing residues from the cluster model on protein scaffolds by matching the atoms of residues with atoms of the protein scaffold. The backbone coordinates of the pocket residues are also modified in the process. The qualities of the generated pockets are evaluated and ranked based on a statistical scoring function. We take the top 100 designed pockets for evaluation. The output of DEPACT+PACMatch is complete protein structures with redesigned pockets. In the paper, we only use DEPACT to represent the whole method of DEPACT+PACMatch for conciseness.

**RFDiffusionAA** [48] [4] is the latest version of RFDiffusion which combines a residue-based representation of amino acids and atomic representations of all other groups to model protein-small molecules/metals/nucleic acids/covalent modification complexes. Starting from random distributions of amino acid residues surrounding target small molecules, RFDiffusionAA can directly generate the small molecule binding protein backbone. Furthermore, with LigandMPNN [22], the latest version of ProteinMPNN[21], we can assign residue types and predict sidechain conformations considering the protein-ligand interactions. Experiments in RFDiffusionAA [48] show that the generated protein by RFDiffusionAA has better binding affinity than those obtained by RFDiffusion with auxiliary potential. We use the provided checkpoints of RFDiffusionAA for all the experiments since the training code is unavailable.

**dyMEAN** [47] [5] is an end-to-end full-atom model for E(3)-equivariant antibody design given the epitope and the incomplete sequence of the antibody. Its previous version, MEAN [46], only considers the backbone atoms, while dyMEAN considers the complete atom structure and performs better on downstream tasks. Generally, dyMEAN co-designs antibody sequence and structure via a multi-round progressive full-shot refinement manner, which is more efficient than auto-regressive or diffusion-based approaches. An adaptive multi-channel equivariant encoder is used in dyMEAN, which can process protein residues of variable sizes when considering full atoms. To adapt dyMEAN to our pocket design task, we replace the antigen with the target ligand molecule to provide the context information for pocket generation. We set the hidden size as 128, the number of layers as 3, and the number of iterations for decoding as 3.

**FAIR** [92] [6] is our previous method for full atom pocket sequence-structure co-design. FAIR operates in two steps, proceeding in a coarse-to-fine manner (backbone refinement to full atoms refinement, including side chains) for full-atom generation. In FAIR, residue types and atom coordinates are updated using a hierarchical graph transformer composed of a residue-level and atom-level encoder. The number of layers for the atom and residue-level encoder are 6 and 2, respectively. $K_a$ and $K_r$ are set as 24 and 8 respectively. The number of attention heads is set as 4; The hidden dimension $d$ is set as 128.

---

[3]https://github.com/chenyaoxi/DEPACT_PACMatch
[4]https://github.com/baker-laboratory/rf_diffusion_all_atom
[5]https://github.com/THUNLP-MT/dyMEAN
[6]https://github.com/zaixizhang/FAIR

