# OpenReview forum: "Generalized Protein Pocket Generation with Prior-Informed Flow Matching"
_NeurIPS.cc/2024/Conference — NeurIPS 2024 spotlight_

### Official Review · Reviewer_bfsA · 2024-07-06

**Soundness:** 2
**Presentation:** 3
**Contribution:** 3
**Rating:** 8
**Confidence:** 4

**Summary:**

In this paper, the authors proposed PocketFlow, a protein-ligand interaction prior-informed flow matching model for protein pocket generation. The flow matching for backbone frames, sidechain torsion angles, and residue/interaction types are appropriately defined. To enhance the structural validity and binding affinity of the generated pockets, the authors proposed to leverage affinity and geometry guidance in the sampling process. Extensive experiments show that PocketFlow provides a generalizable model for pocket generation covering various ligand modalities such as small molecules, nucleic acids, and peptides.

**Strengths:**

1. PocketFlow leverages domain knowledge including protein-ligand interactions and geometric constraints to enhance pocket generation quality, which is quite novel.
2. PocketFlow extends the pocket generation task to broad ligand modalities such as small molecules, RNA, and peptides. Experiments show that the prior guidance can effectively improve the generalization capability.
3. In experiments, the advantage of PocketFlow over state-of-the-art baselines such as RFDiffuisionAA is obvious, achieving an average improvement of 1.29 in Vina score and 0.05 in scRMSD in Table 1.
4. The description of the PocketFlow algorithm is quite clear, and a preliminary version of code is provided for reproducibility.
5. The authors conducted comprehensive ablation studies to show the contribution of affinity/geometry guidance and protein-ligand interaction learning.

**Weaknesses:**

1. The backbone model of PocketFlow is modified from existing work such as FrameDiff and is less novel.
2. In lines 252-253, the authors said PocketFlow is only pretrained on protein-small molecule datasets, i.e., CrossDocked and Binding MOAD. Is it possible to train PocketFlow on the combination of protein-small molecule/peptide/RNA datasets for better performance?
3. In experiments on protein-peptide and RNA, PocketFlow represents peptide/RNA ligands as molecules. Could the frame representation for the protein residues also applied to peptide/RNA ligands?

**Questions:**

Please see the Weaknesses.

**Limitations:**

The limitations and broader impacts are well discussed in Section 5.5 of the main paper.

---

> ### Author Rebuttal · Authors · 2024-08-05
>
> We thank the reviewer for the constructive comments and appreciation!
>
> **Comment 1**: The backbone model of PocketFlow is modified from existing work such as FrameDiff and is less novel.
>
> **Response 1**: Thanks for the question! This paper is an application-driven paper and would be of great interest to the growing AI for Science community. To well model the protein-ligand complex, we appropriately define **multi-modal flow matching** process for different components, including SE(3) flow matching for the protein backbone, torsional flow matching for sidechain torsion angles, and categorical flow matching for residue Types and interaction types. The incorporation of additional domain constraints into the framework is also not straightforward. For example, we novelly **formulate the complicated geometrical constraints (see Appendix. B) into guidance terms** for flow matching. To **tackle the non-differentiability of residue type sampling**, we propose the Sidechain Ensemble technique for the interaction geometry calculation. We believe the above-mentioned techniques and practices will also inspire the machine learning research community.
>
> **Comment 2**: In lines 252-253, the authors said PocketFlow is only pretrained on protein-small molecule datasets, i.e., CrossDocked and Binding MOAD. Is it possible to train PocketFlow on the combination of protein-small molecule/peptide/RNA datasets for better performance?
>
> **Response 2**: Thanks for the detailed question! In Table 2, we explore whether the pretrained PocketFlow on the combination of CrossDocked and Binding MOAD can generalize to peptide and RNA-binding pocket design. In the rebuttal period, we further finetuned the pretrained model on protein-peptide/RNA datasets constructed following previous works [1-2]. The datasets are split based on peptide/RNA sequence similarity. In the following table, we report the latest results and observe that finetuning PocketFlow achieves better results across most of the metrics compared to the original ones in the submission. We will include the new experiments and discussions in our revised paper.
>
> | Model         | PPDBench              |                   |                  | PDBBind RNA          |                   |                  |
> |---------------|-----------------------|-------------------|------------------|----------------------|-------------------|------------------|
> |               | AAR (↑)               | scRMSD (↓)        | ΔΔG (↓)          | AAR (↑)              | scRMSD (↓)        | ΔΔG (↓)          |
> | Test set      | -                     | 0.64              | -                | -                    | 0.59              | -                |
> | dyMEAN        | 26.29±1.05%           | 0.71±0.05         | -0.23±0.04       | 25.90±1.22%          | 0.71±0.04         | -0.18±0.03       |
> | FAIR          | 32.53±0.89%           | 0.86±0.04         | 0.05±0.07        | 24.90±0.92%          | 0.80±0.05         | 0.13±0.05        |
> | RFDiffusionAA | *46.85±1.45%*           | **0.65±0.06**         | *-0.62±0.05*       | *44.69±1.90%*          | **0.65±0.03**         | *-0.45±0.07*       |
> | PocketFlow    | **48.54±1.37%**       | *0.67±0.03*        | **-1.15±0.07**   | **46.31±1.22%**      | *0.68±0.02*         | **-0.90±0.04**   |
>
> - **Bold**: Best results
> - *Italic*: Second best
>
> [1] Li J, Cheng C, Wu Z, et al. Full-Atom Peptide Design based on Multi-modal Flow Matching[J]. ICML, 2024.
> [2] Nori D, Jin W. RNAFlow: RNA Structure & Sequence Design via Inverse Folding-Based Flow Matching[J]. ICML, 2024.
>
> **Comment 3**: In experiments on protein-peptide and RNA, PocketFlow represents peptide/RNA ligands as molecules. Could the frame representation for the protein residues also applied to peptide/RNA ligands?
>
> **Response 3**: Thanks for the insightful question! Yes, the peptide and RNA structures can also be modeled as frames and there are some recent reference papers such [1-3]. In PocketFlow, we represent peptide/RNA ligands as molecules for simplicity and explore the generalization capability of PocketFlow on other ligand domains. We will include the discussions of representation in our revised paper.
>
> [3] Anand R, Joshi C K, Morehead A, et al. RNA-FrameFlow for de novo 3D RNA Backbone Design[C]//ICML 2024 Workshop on Structured Probabilistic Inference {\&} Generative Modeling.

---

> ### Comment · Reviewer_bfsA · 2024-08-13
> **Response to Author**
>
> Hi,
>
> Thanks for the response, my concerns have been successfully addressed!
>
> Regards,

---

### Official Review · Reviewer_e3xX · 2024-07-08

**Soundness:** 3
**Presentation:** 3
**Contribution:** 3
**Rating:** 7
**Confidence:** 4

**Summary:**

This paper studies the task of generalized ligand-binding protein pocket generation. To tackle the challenges of existing works, the authors proposed PocketFlow, a generative model that incorporates protein-ligand interaction priors based on flow matching. PocketFlow explicitly learns the protein-ligand interactions during training and leverages multi-granularity guidance to generate high-quality pockets during sampling. Experiments show that PocketFlow is a generalized generative model across various ligand modalities, including small molecules, peptides, and RNA.

**Strengths:**

1. The paper is well-written and easy to follow. The illustration of Figure 1 is attractive. The technical details and the code are provided for better reproducibility.
2. PocketFlow represents the first few works to broaden the scope of pocket generation tasks to include various ligand modalities, such as small molecules, nucleic acids, and peptides.
3. PocketFlow effectively combines the latest flow-matching models with prior knowledge (affinity guidance and interaction geometry guidance) to generate protein pockets with enhanced affinity and validity.
4. The proposed algorithms overall sound valid to me. The tasks and evaluation metrics of peptide/RNA-binding pocket design are also well formulated.
5. Experiments show that PocketFlow achieves an average improvement of 1.29 in Vina score and 0.05 in scRMSD. The authors also leverage PoseCheck to evaluate the protein-ligand interactions and steric clashes, which makes the evaluation comprehensive.

**Weaknesses:**

1. What is the time cost of PocketFlow and baseline methods in generating protein pockets? Would the prior knowledge-based guidance bring an extra burden to pocket generation?
2. In line 222, the authors mentioned sidechain ensemble for the interaction geometry calculation, which is too concise. The authors are recommended to elaborate more on this in the main pages.
3. In Table 2, some metrics of PocketFlow are not the best and RFDiffusionAA has better results.

**Questions:**

1. Why didn’t the authors apply discrete flow matching methods for residue/interaction type prediction, e.g., [1]?

[1] Dirichlet flow matching with applications to dna sequence design, arXiv preprint arXiv:2402.05841, 2024

**Limitations:**

The authors discussed the limitations and broader impacts in Section 5.5.

---

> ### Author Rebuttal · Authors · 2024-08-05
>
> We thank the reviewer for the valuable comments and appreciation!
>
> **Comment 1**: What is the time cost of PocketFlow and baseline methods in generating protein pockets? Would the prior knowledge-based guidance bring an extra burden to pocket generation?
>
> **Response 1**: Thanks for the detailed question! In Figure 7 of the submitted paper, we compare the average generation time of different models. Firstly, we observed that PocketFlow is much more efficient than template-matching methods (DEPACT) and diffusion-based models (RFDiffuisonAA). The generation time is only larger than dyMEAN and FAIR which are based on equivariant translation. Considering the performance improvement brought by PocketFlow (1.29 in Vina Score and 0.05 in scRMSD), the additional overhead is acceptable. We also compared PocketFlow with its variants without guidance and observed that the guidance mechanisms are quite efficient and would not introduce much overhead. We will add more clear discussions in our revised paper.
>
>
> **Comment 2**: In line 222, the authors mentioned sidechain ensemble for the interaction geometry calculation, which is too concise. The authors are recommended to elaborate more on this in the main pages.
>
> **Response 2**: Thanks for the constructive suggestion! Due to the page limits, we include the detailed elaborations of sidechain ensemble technique in the appendix. PocketFlow takes the co-design scheme, where the residue type/side chain structure of the pocket is not determined during sampling. Directly sampling from the residue type distribution makes the model not differentiable. We propose to use the sidechain ensemble for the interaction geometry calculation, i.e., the weighted sum of geometric guidance with respect to residue types. We will put the detailed descriptions of sidechain ensemble in our final version.
>
> **Comment 3**: In Table 2, some metrics of PocketFlow are not the best and RFDiffusionAA has better results
>
> **Response 3**: Thanks for the detailed comments! We admit that RFDiffusionAA is indeed a strong baseline. In Table 2, we evaluate different methods on peptide/RNA-conditioned protein pocket generation to explore the generalization capability. The performance of PocketFlow is comparable to the state-of-the-art model RFDiffusionAA. Even though some results are not the best, PocketFlow achieves the second best. Such results are quite promising and show the potential of our multi-modality flow matching architecture and domain knowledge-based guidance.
> During rebuttal, we also tried enlarging the pertaining dataset (response 2 to reviewer bfsA) or leveraging SOTA discrete flow matching method (response 4) to improve the performance and achieved promising results.
> In the future, we will keep updating our model and improve its performance.
>
> **Comment 4**: Why didn’t the authors apply discrete flow matching methods for residue/interaction type prediction, e.g., [1]?
>
> [1] Dirichlet flow matching with applications to DNA sequence design, arXiv preprint arXiv:2402.05841, 2024
>
> **Response 4**: Thanks for mentioning the latest discrete flow matching methods, which inspires us a lot. Such discrete flow matching models are perpendicular to our work and can be seamlessly integrated into PocketFlow. During rebuttal, we performed additional experiments and observed promising results of incorporating the discrete flow matching model. We will include the new results with discrete flow matching model in our revised version.
>
> | Model         | CrossDocked           |                   |                  | Binding MOAD         |                   |                  |
> |---------------|-----------------------|-------------------|------------------|----------------------|-------------------|------------------|
> |               | AAR (↑)               | scRMSD (↓)        | Vina (↓)         | AAR (↑)              | scRMSD (↓)        | Vina (↓)         |
> | Test set      | -                     | 0.65              | -7.016           | -                    | 0.67              | -8.076           |
> | DEPACT        | 31.52±3.26%           | 0.73±0.06         | -6.632±0.18      | 35.30±2.19%          | 0.77±0.08         | -7.571±0.15      |
> | dyMEAN        | 38.71±2.16%           | 0.79±0.09         | -6.855±0.06      | 41.22±1.40%          | 0.80±0.12         | -7.675±0.09      |
> | FAIR          | 40.16±1.17%           | 0.75±0.03         | -7.015±0.12      | 43.68±0.92%          | 0.72±0.04         | -7.930±0.15      |
> | RFDiffusionAA | 50.85±1.85%           | 0.68±0.07         | -7.012±0.09      | 49.09±2.49%          | 0.70±0.04         | -8.020±0.11      |
> | PocketFlow    | *52.19±1.34%*       | *0.67±0.04*         | *-8.236±0.16*  | **54.30±1.70%**      | *0.68±0.03*         | **-9.370±0.24**  |
> | w/ discrete flow | **53.87±1.20%**           | **0.66±0.05**         | **-8.310±0.17**      | *53.39±1.65%*          | **0.65±0.03**         | *-9.267±0.31*      |
>
> - **Bold**: Best results
> - *Italic*: Second best

---

> > ### Comment · Reviewer_e3xX · 2024-08-09
> > **Response to author**
> >
> > Thanks for the authors response. After reading the rebuttal, my concerns are resolved and I decided to rise my score.

---

### Official Review · Reviewer_gVL4 · 2024-07-10

**Soundness:** 3
**Presentation:** 3
**Contribution:** 3
**Rating:** 7
**Confidence:** 4

**Summary:**

The paper explores methods for generating protein pockets given a ligand using a flow matching generative approach. Unlike previous methods, the proposed approach integrates additional constraints into the flow matching learning process to guide the search for relevant pockets. Two types of constraints are considered:

1. Affinity score prediction, which utilizes an oracle predictive model to guide the generation process.
2. Geometric constraints, which ensure that the distances between atoms involved in specific types of bonds remain below certain thresholds.

The authors examine the effectiveness of these constraints, incorporating both domain knowledge and transferred knowledge from an affinity prediction model, in the generation of protein pockets. They benchmark their approach against state-of-the-art methods that do not use such constraints. The results demonstrate that the new pockets generated have improved RMSD and Vina scores.

**Strengths:**

The concept of incorporating additional constraints into Flow Matching is interesting. While conditional flow matching exists, determining the appropriate constraints for pocket generation requires substantial domain knowledge and engineering effort to be effective.

The results are promising, as the authors conducted numerous experiments and comparisons with state-of-the-art methods. It is evident that considerable time was invested in gathering results from these methods.

The paper is well-written and enjoyable to read, even for those with limited knowledge of pocket generation problems.

**Weaknesses:**

I have a concern about the inclusion of the affinity oracle predictor, which was trained on separate datasets. The authors did not specify which dataset was used to train their predictor or address the potential for information leakage between the training dataset and the test set used for pocket finding. Since the predictor provides the flow matching with a prior on which pocket might be relevant for a given ligand, any leakage would give a clear advantage. I looked for more detailed information about the affinity predictor in the appendix but found none regarding the training dataset.

I also question the novelty of the work in terms of its technical contribution to the machine learning community. While conditional flow matching itself is not new, the novel aspect of this work is the incorporation of additional domain constraints into the framework. Therefore, it is unlikely to be of significant interest to the machine learning research community from a methodological development perspective.

Additionally, since the authors retrained the baseline models with additional data, it is important that they report how hyperparameters were selected and tuned for these baselines.

**Questions:**

1. Could you please address the potential leakage between the dataset used for training the affinity predictor and the datasets used for assessing the pocket generation?
2. Could you please clarify the impacts of HPO of the baselines and the proposed method in this work?

**Limitations:**

The authors have addressed relevant limitations of the work in the paper.

---

> ### Author Rebuttal · Authors · 2024-08-05
>
> We thank the reviewer for the valuable comments! Our replies are listed below:
>
> **Comment 1**: The authors did not specify which dataset was used to train their predictor or address the potential for information leakage between the training dataset and the test set used for pocket finding.
>
> **Response 1**: In the original submission lines 739-744, we described the dataset used to train the predictors: To train the binding affinity predictor, we first annotate the data points in the corresponding training set: data points are annotated 1 if their affinity is higher than the average score of the dataset, otherwise 0.
>
> **Therefore, no additional datasets are used and there is no data leakage risks**. In the revised paper, we will highlight the training dataset for the predictors and address the potential concerns.
>
> **Comment 2**: I also question the novelty of the work in terms of its technical contribution to the machine learning community. While conditional flow matching itself is not new, the novel aspect of this work is the incorporation of additional domain constraints into the framework. Therefore, it is unlikely to be of significant interest to the machine learning research community from a methodological development perspective.
>
> **Response 2**: Thanks for the comment! This paper is an application-driven paper and would firstly be of great interest to the growing AI for Science community. To well model the protein-ligand complex, we appropriately define **multi-modal flow matching process for different components**, including SE(3) flow matching for the protein backbone, torsional flow matching for sidechain torsion angles, and categorical flow matching for residue Types and interaction types. The incorporation of additional domain constraints into the framework is also not straightforward. For example, we novelly **formulate the complicated geometrical constraints (see Appendix. B) into guidance terms** for flow matching. To **tackle the non-differentiability of residue type sampling**, we propose the Sidechain Ensemble technique for the interaction geometry calculation. **We believe the above-mentioned techniques and practices will also inspire the machine learning research community.**
>
> **Comment 3**: since the authors retrained the baseline models with additional data, it is important that they report how hyperparameters were selected and tuned for these baselines.
>
> **Response 3**: In Appendix. G Baseline Implementation, we described the details of running baseline methods. DEPACT is a template-matching method. **We used the recommended hyperparameters such as the weights of scoring functions from the original paper.** RFDiffusionAA is the state-of-the-art diffusion model for generalized biomolecular modeling and generation. **We use the provided checkpoints and the recommended hyperparameter setting from the paper** because the training code and data are not available. dyMEAN and FAIR are end-to-end deep generative models for protein sequence-structure codesign. **We performed a grid search over the key hyperparameters such as the number of layers and iterations based on the performance of validation datasets (Vina score).** Finally, we set the hidden size as 128, the number of layers as 3, and the number of iterations for decoding as 3 for dyMEAN. For FAIR, The number of layers for the atom and residue-level encoder are 6 and 2, respectively. Ka and Kr are set as 24 and 8 respectively. The number of attention heads is set as 4; The hidden dimension d is set as 128.
>
> We will describe the selection/tuning of hyperparameters of baseline more clearly in the revised paper.

---

> > ### Comment · Reviewer_gVL4 · 2024-08-13
> >
> > Thank you for your response to my concern, I read the paragraph in the appendix:
> > "To train the binding affinity predictor, we first annotate the data points in the corresponding training
> > set: data points are annotated 1 if their affinity is higher than the average score of the dataset, otherwise 0."
> > Could you please clarify more on "higher than the average score of the dataset", the average score you mentioned in this sentence is calculated on the training data?

---

> > > ### Author Response · Authors · 2024-08-13
> > > **Thanks for the response!**
> > >
> > > Yes, the average score (Vina score) is calculated on the training data. We annotate the data point as 1 if its calculated Vina score is lower than the average (higher affinity); we annotate the data point as 0 if its calculated Vina score is higher than the average (lower affinity). Therefore, all the calculations are based on the training data and there are no data leakage risks. We will make the statements in the paper clearer in the revised version. Thanks for the comments!
> > >
> > > Bests,
> > > Authors

---

> > > > ### Comment · Reviewer_gVL4 · 2024-08-13
> > > >
> > > > Thank you for clarification, I increase my score.

---

### Official Review · Reviewer_gNH7 · 2024-07-23

**Soundness:** 3
**Presentation:** 3
**Contribution:** 3
**Rating:** 6
**Confidence:** 4

**Summary:**

The paper proposed PocketFlow, a generative model for designing protein pockets that bind with ligands. It aims to overcome limitations in existing methods by incorporating protein-ligand interaction priors and utilizing flow matching. PocketFlow is designed to handle multiple ligand modalities and demonstrates superior performance on various benchmarks.

**Strengths:**

- The paper provides detailed methodology and includes anonymous code to reproduce the results.
- The proposed method generalizes across various ligand modalities, including small molecules, peptides, and RNA.
- The model outperforms existing methods on multiple benchmarks, demonstrating significant improvements in Vina scores and scRMSD.
- The model explicitly models key protein-ligand interactions, enhancing binding stability and affinity.

**Weaknesses:**

- The method only considers interactions between protein and ligand, potentially neglecting interactions between protein sidechains within the pocket region.
- For some residue types, there might be π instead of 2π symmetry in the sidechain structures, which the proposed method seems to simplify.
- The use of flow matching and multiple guidance mechanisms could result in higher computational costs compared to simpler models.

**Questions:**

1. Why were these four types of interactions (hydrogen bond, salt bridge, hydrophobic, π-π stacking) chosen? Are there other interactions existing in protein-ligand binding that should be considered?
2. Can the proposed method be extended to protein-ligand docking (fixing the pocket type and structure)?
3. What is the efficiency of calculating different interaction types?
4. Is there any metric to evaluate the correctness of the interactions generated?
5. The original IPA relies on the rotational equivariance of frame orientation to achieve model’s invariance. However, the proposed method additionally treats the ligand atom as a residue and uses an invariant identity matrix to represent its orientation. Will the proposed IPA still output invariant embeddings?
6. How are the guidance coefficients for different guidance mechanisms determined?
7. Is RFDiffusionAA trained on the same dataset as the proposed model?

---

> ### Author Rebuttal · Authors · 2024-08-05
>
> We thank the reviewer for the constructive comments and appreciation!
>
> **Comment 1**: The method only considers interactions between protein and ligand, potentially neglecting interactions between protein sidechains within the pocket region.
>
> **Response 1**: Thanks for the insightful comment! In PocketFlow, we explicitly consider protein-ligand interactions as guidance terms because protein-ligand interactions mainly contribute to the protein-ligand binding affinity. They are also learned implicitly within our model architecture: the pairwise attentions capture inter-residue interactions and the supervision of the predicted sidechain torsion angles encourages forming valid and stable protein conformations.
>
> We note that **PocketFlow has a flexible framework and can be generalized to model protein sidechain interactions with minor modifications**. For model simplicity and computational efficiency, we only consider the protein-ligand interactions in our current version of PocketFlow. We will include the above discussions in our revised paper.
>
> **Comment 2**: For some residue types, there might be π instead of 2π symmetry in the sidechain structures, which the proposed method seems to simplify.
>
> **Response 2**: Thanks for the constructive comments! Yes, we are aware some sidechain torsion angles are 180◦-rotation-symmetric, such that the predicted torsion angle χ and χ + π result in the same physical structure. Since only 4 of 39 possible sidechain torsion angles have such π symmetric properties, we did not consider them for simplicity in our preliminary version of PocketFlow. Following AlphaFold2 (supplementary 1.9.1), we can include such π symmetric constraints by providing the alternative ground truth torsion angle for training with minor modifications of the code. We will include the discussions and the implementation in our revised paper.
>
> **Comment 3**: The use of flow matching and multiple guidance mechanisms could result in higher computational costs compared to simpler models.
>
> **Response 3**: Thanks for the question! In Figure 7 of the submitted paper, we compare the average generation time of different models. Firstly, we observed that PocketFlow is much efficient than template-matching methods (DEPACT) and diffusion-based models (RFDiffuisonAA). The generation time is only larger than dyMEAN and FAIR. Considering the performance improvement brought by PocketFlow (1.29 in Vina Score and 0.05 in scRMSD), the additional overhead is acceptable. We also compare PocketFlow with its variants without guidance and observed that the guidance mechanisms are quite efficient and would not introduce much overhead.
>
> **Comment 4**: Why were these four types of interactions chosen? Are there other interactions existing in protein-ligand binding that should be considered?
>
> **Response 4**: In PocketFlow, the four types of interactions are chosen because they are the most frequently encountered and are crucial for strong binding stability and affinity. Previous works, e.g., KGDiff ([93] in the paper) also consider the four interactions and managed to improve binding affinity and generation quality. In Appendix. B, we described the details of the four dominant interactions. There are some other interactions in protein-ligand binding such as Van der Waals Forces and Metal Coordination. We did not consider them due to their weak contributions to binding affinity or low occurring frequency. Moreover, we only consider the four dominant interaction types for computation efficiency.
>
> **Comment 5**: Can PocketFlow be extended to protein-ligand docking (fixing the pocket type and structure)?
>
> **Response 5**: Yes, PocketFlow has a generalized and robust architecture that can be extended to protein-ligand docking by e.g., fixing the pocket type and structure. During the rebuttal period, we conducted preliminary experiments on adapting PocketFlow to the blind docking tasks. We conduct experiments on the PDBbind v2020 dataset and follow previous works such as DiffDock and FABind for the data preprocessing. For our testing phase, we utilized 363 complexes recorded after 2019. In the table below, we follow previous works and report the percentage of ligands RMSD below 2/5Å and the centroid distance below 2/5Å. We observe that even though PocketGen is not specially optimized for docking, it still achieves strong performance compared with state-of-the-art baselines.
>
> | Methods       | Ligand RMSD       |                 | Centroid Distance   |                 |  |
> |---------------|--------------------|-----------------|---------------------|-----------------|---------------------|
> |               | % Below 2Å | % Below 5Å |                 | % Below 2Å | % Below 5Å |                     |
> | QVINA-W       | 15.3     | 39.5     |                 | 25.7     | 39.5     | 49*                |
> | GNINA         | 12.5     | 37.0     |                 | 20.4     | 37.9     | 146                |
> | SMINA         | 13.5     | 39.1     |                 | 29.9     | 41.7     | 146*               |
> | GLIDE         | 19.6     | 32.2     |                 | 35.4     | 46.4     | 1405*              |
> | VINA          | 10.3     | 27.3     |                 | 20.4     | 37.3     | 205*               |
> | EQUiBind      | 3.4      | 43.8     |                 | 16.7     | 43.8     | **0.03**           |
> | TANKBind      | 4.3      | 44.8     |                 | **44.0**     | 70.8     | 0.87               |
> | E3Bind        | 4.5      | 34.3     |                 | 33.8     | 66.0     | 0.83               |
> | DiffDock     | **32.0**     | 48.3     |                 | 33.8     | 62.6     | 20.83              |
> | FABind        | 19.4     | *64.0*      |                 | 5.9      | **75.7**     | *0.12*                |
> | PocketFlow       | *30.2*     | **65.7**     |                 | *38.0*      | *71.3*    | 0.45               |
>
> - **Bold**: Best results
> - *Italic*: Second best

---

> ### Author Response · Authors · 2024-08-05
> **Further Response to Reviewer gNH7**
>
> **Comment 6**: What is the efficiency of calculating different interaction types?
>
> **Response 6**: Thanks for the question! In Figure 7 included in the Appendix, we compared the average generation time of PocketFlow and its variants such as PocketFlow without different guidance terms. We can observe that calculating the protein-ligand interaction for guidance will not bring much overhead (~28% of the total generation time).
>
> To detect the different interaction types in the generated pockets, we leverage PLIP and posecheck. They are also efficient tools and it totally takes around 20 seconds to process 100 generated pockets.
>
> **Comment 7**: Is there any metric to evaluate the correctness of the interactions generated?
>
> **Response 7**: Thanks for the question! In PocketFlow, we used the protein-ligand interaction profiler (PLIP) [69] to detect and annotate the protein-ligand interactions for each residue by analyzing their binding structure (Table 3 and lines 304-314 of the paper). Generally, PLIP is based on physical/chemical rules and employs four steps including structure preparation, functional characterization, rule-based matching, and filtering to detect the generated interactions. In the PLIP paper [69], the authors compared the detected/ground truth interactions of 30 literature-validated examples and achieved good consistency. Therefore, we can regard the detected interactions by PLIP as correct.
>
> **Comment 8**: The original IPA relies on the rotational equivariance of frame orientation to achieve model’s invariance. However, the proposed method additionally treats the ligand atom as a residue and uses an invariant identity matrix to represent its orientation. Will the proposed IPA still output invariant embeddings?
>
> **Response 8**: Thanks for the insightful question! We agree that IPA relies on the rotational equivariance of frame orientation to achieve the model’s invariance. In our case, there is no canonical orientation for the ligand atoms and we set them as an identity matrix for simplicity.
> To achieve invariant embeddings, we can initialize the protein scaffold by aligning the protein principal axes with the coordinate axes. This can be achieved by subtracting the center of mass (COM), computing the inertia matrix, diagonalizing the inertia matrix, and aligning the protein to principal axes. In experiments below, we compared such an initialization strategy with the default setting (only subtracting COM). The experimental results are comparable and show no significant differences. Therefore, we use the default setting for simplicity. We will include more discussions in our revised paper.
>
> | Model         | CrossDocked           |                   |                  | Binding MOAD         |                   |                  |
> |---------------|-----------------------|-------------------|------------------|----------------------|-------------------|------------------|
> |               | AAR (↑)               | scRMSD (↓)        | Vina (↓)         | AAR (↑)              | scRMSD (↓)        | Vina (↓)         |
> | PocketFlow    | 52.19±1.34%       | 0.67±0.04         | -8.236±0.16 | 54.30±1.70%      | 0.68±0.03         | -9.370±0.24  |
> | w/ principal axes aligning | 52.12±1.29%          | 0.67±0.03        | -8.227±0.18      | 54.47±1.73%          | 0.69±0.03         | -9.362±0.28     |
>
> **Comment 9**: How are the guidance coefficients for different guidance mechanisms determined?
>
> **Response 9**: In the default setting, the guidance coefficients of PocketFlow including $\gamma, \xi_1, \xi_2,$ and $\xi_3$ are set as 1 and achieve good results. We also explore the influence of guidance coefficients in the appendix. For example, in Figure. 8 of the submitted paper, we explore the impact of Affinity Guidance Strength ($\gamma$) on various generation metrics. As $\gamma$ is scaled up, the Vina Score significantly improves and quickly stabilizes; AAR initially increases before gradually decreasing; scRMSD, on the other hand, increases with higher $\gamma$. These observations underscore the importance of selecting an appropriate $\gamma$ to effectively balance the guidance and unconditional terms. While Affinity Guidance promotes the generation of high-affinity pockets, an excessively high $\gamma$ can result in less valid pocket sequences or structures. In the default configuration, $\gamma$ is set to 1 for simplicity.
>
> **Comment 10**: Is RFDiffusionAA trained on the same dataset as the proposed model?
>
> **Response 10**: As indicated in lines 822-823, we used the provided checkpoints of RFDiffusionAA for all the experiments because the training code is not available (https://github.com/baker-laboratory/rf_diffusion_all_atom). The original training data of RFDiffusionAA contains 121,800 protein-small molecule structures, 112,546 protein-metal complexes, and 12,689 structures with covalently modified amino acids, which represent a broad set of protein-ligand complex structures.

---

### Decision · Program_Chairs · 2024-09-25

**Decision:**

Accept (spotlight)

**Comment:**

The authors present a novel protein-ligand interaction prior flow-matching model for protein pocket generation. The approach defines flow matching for backbone frameworks, side-chain torsion angles, and residue/interaction types. To enhance the structural validity and binding affinity of generated pockets, the authors propose incorporating affinity and geometric guidance during the sampling process. Experimental results show that PocketFlow is a generalized generative model capable of handling a wide range of ligand types, including small molecules, peptides, and RNA, achieving SOTA performance.

The reviewers universally found this work interesting and timely, particularly given the growing ability to protein design. Despite some minor concerns, this is a strong paper that I recommend for acceptance to NeurIPS.